# *Thiocapsa*, *Lutimaribacter*, and *Delftia* Are Major Bacterial Taxa Facilitating the Coupling of Sulfur Oxidation and Nutrient Recycling in the Sulfide-Rich Isinuka Spring in South Africa

**DOI:** 10.3390/biology14050503

**Published:** 2025-05-05

**Authors:** Henry Joseph Oduor Ogola, Ramganesh Selvarajan, Somandla Ncube, Lawrence Madikizela

**Affiliations:** 1Department of Environmental Sciences, College of Agriculture and Environmental Sciences (CAES), University of South Africa, Florida Science Campus, Johannesburg 1710, South Africa; 2Institute of Deep-Sea Science and Engineering (IDSSE), Chinese Academy of Sciences (CAS), Sanya 572099, China; 3Department of Chemistry, Durban University of Technology, P.O. Box 1334, Durban 4000, South Africa; somandlan@dut.ac.za; 4Institute of Nanotechnology and Sustainability (iNanos), University of South Africa, Florida Science Campus, Johannesburg 1710, South Africa; madiklm@unisa.ac.za

**Keywords:** sulfur cycling, Isinuka sulfur pool, microbiome analysis, *Thiocapsa*, 16S rRNA gene sequencing

## Abstract

Sulfur is a critical element in natural ecosystems, playing a central role in energy transfer and nutrient cycling. Despite its importance, the way microbes drive sulfur transformations in unique environments like the Isinuka sulfur pool, known for its healing properties, is not well understood. This study aimed to explore the microbial communities in the water and sediments of this sulfur-rich pool, focusing on their diversity, roles, and impact on sulfur and nutrient cycling. By using advanced DNA sequencing technologies, we discovered distinct microbial populations in the water and sediments. The anoxic, light-exposed water column fostered tightly coupled sulfur, carbon, and nitrogen cycling, driven by specialized microbial consortia adapted to anaerobic phototrophs *Thiocapsa* and *Delftia*, while sediments hosted sulfate-reducing microbes in the family Desulfobacteraceae, adapted to low-oxygen environments. These bacteria work together to maintain sulfur cycling, which is critical for sustaining life in the ecosystem. Notably, a high abundance of unclassified Coxiellaceae was detected in the sediments, which may be linked to human contamination due to the pool’s use for balneotherapy. The study also revealed that these microbes contribute to broader processes like nitrogen and carbon cycling, impacting greenhouse gas regulation. Understanding these mechanisms not only enhances scientific knowledge, but could also inform environmental management strategies and the development of sustainable technologies. This research highlights the importance of preserving unique ecosystems like Isinuka for their ecological value and potential societal benefits.

## 1. Introduction

Mesophilic sulfide-rich springs represent geochemically distinct environments characterized by elevated concentrations of sulfur compounds, steep redox gradients, and often extreme physicochemical conditions [1,2]. These systems contain elevated levels of reduced sulfur species, such as hydrogen sulfide (H_2_S), elemental sulfur (S⁰), and sulfite (SO_3_^2−^), along with oxidized forms like sulfate (SO_4_^2−^), which together fuel a continuous sulfur cycle driven by microbial activity [3,4]. The pronounced redox gradients, resulting from the stratification of oxygen and sulfur compounds, create discrete ecological niches ranging from oxygen-rich photic zones at the surface to anoxic, sulfur-dense environments at depth [1]. This stratification supports a metabolically diverse microbial community, including sulfur-oxidizing and sulfate-reducing bacteria, which exploit these gradients for energy production and nutrient cycling [1,4].

An excellent example of such sulfur-rich, geochemically distinct environments is the Isinuka Mud Cave and Sulfur Pool, located in the OR Tambo District Municipality, Eastern Cape, South Africa, which displays a range of unique geochemical and ecological attributes. Situated ~800 m above mean sea level and approximately 20 km west of Port St. Johns, the site is defined by elevated concentrations of total dissolved solids (TDS), turbidity, chloride (Cl^−^), and ammonium nitrogen (NH_4_^+^-N), surpassing acceptable thresholds for potable waters [5,6,7]. The spring waters are moderately hard and salty, and exhibit a neutral to slightly alkaline pH ranging from 6.87 to 8.33. A pervasive hydrogen sulfide (H_2_S) odor, for which the site is named “Isinuka”—meaning “place of smell” in the local Mpondo language—pervades the environment, further indicating sulfur enrichment.

The unique geochemical characteristics of the Isinuka sulfur pools, such as the turbid, dark grey sediments (mud), the striking pink hue of the sulfur pool linked to enriched phototrophic purple sulfur bacteria (PSB), and the presence of bubbling H_2_S, are indicative of active microbial sulfur cycling, a process that not only shapes the environment, but also sustains the site’s cultural and ecological significance [5,6,7]. These features suggest a dynamic sulfur cycling process driven by a specialized microbial community capable of thriving under fluctuating redox conditions. Such microbial sulfur cycling may be integral to maintaining the system’s geochemical equilibrium, while supporting a unique extremophile community adapted to these extreme conditions. Culturally, the Isinuka Springs hold significance for the Mpondo people, who regard the site as sacred, attributing therapeutic properties to the sulfur-rich waters [5,6,7]. The combination of geochemical properties renders Isinuka a rich natural laboratory for studying sulfur biogeochemistry, microbial ecology, and the adaptive mechanisms employed by extremophiles in response to environmental conditions.

Microbial sulfur cycling is a cornerstone of biogeochemical processes in sulfur-rich ecosystems, such as sulfur pools and springs, where sulfur compounds serve both as substrates and products of microbial metabolism. Central to this cycle are key microbial processes, including phototrophic and chemotrophic sulfide oxidation, sulfate reduction, and elemental sulfur transformations [1,4,8]. Phototrophic sulfur-oxidizing bacteria (e.g., *Thiocapsa*, *Chromatium*, and *Chlorobium*) and chemotrophic sulfur oxidizers (e.g., *Thiobacillus*, *Beggiatoa*, and *Acidithiobacillus*) drive the oxidation of H_2_S to SO_4_^2−^ or S⁰, contributing to the system’s sulfur chemistry [2,9]. In contrast, sulfate-reducing bacteria (SRB) from genera like *Desulfovibrio*, *Desulfobacter*, and *Desulfococcus* reduce SO_4_^2−^ to H_2_S under anoxic conditions, completing the sulfur cycle [1,10]. Furthermore, S⁰ serves as a critical intermediate, undergoing transformations catalyzed by bacteria such as *Thiobacillus denitrificans*, *Paracoccus denitrificans*, and *Sulfolobus* species, which either oxidize it to SO_4_^2−^ or reduce SO_4_^2−^ to sulfur [2,3,11,12]. These processes regulate nutrient dynamics, facilitate sediment formation through metal sulfide precipitation, and drive energy flow, supporting primary productivity. In this way, sulfur cycling not only underpins ecosystem stability by maintaining redox gradients, but also promotes microbial diversity, which is essential for the functioning of sulfur-based ecosystems [1,4].

While microbial sulfur cycling is central to the functioning of these ecosystems, the complexity and diversity of microbial communities involved in sulfur transformations—especially within sulfur pools and springs—are still not well understood [1,2]. Previous studies have identified key taxonomic groups, such as SOB and SRB, but the specific microbial interactions and their contributions to sulfur cycling have yet to be fully elucidated. Functional dominance within these communities has often been inferred, rather than directly linked to environmental variables such as redox gradients, temperature, or sulfur compound availability, which limits our understanding of the microbial mechanisms driving sulfur transformations.

The Isinuka sulfur pool, with its extreme physicochemical conditions and dynamic sulfur cycling, presents an ideal environment to address the current gaps in our understanding of microbial sulfur cycling [5,6,7]. Despite its significance, the microbial communities inhabiting the Isinuka sulfur pool remain underexplored. The circumneutral, sulfur-rich, dynamic redox nature of the Isinuka ecosystem likely supports a highly specialized microbial community, yet the structural and functional roles of these microorganisms in sulfur cycling and the cycling of other nutrients remain largely uncharacterized.

This study aimed to address this knowledge gap by identifying the dominant microbial players involved in sulfur cycling and the cycling of other nutrients in the Isinuka sulfur pool. Using high-throughput 16S rRNA amplicon sequencing on an Illumina MiSeq platform, we explored the composition and structure of the bacterial community in sediment and water samples. Furthermore, the predictive function profiling tool PICRUSt2 (Phylogenetic Investigation of Communities by Reconstruction of Unobserved States) was employed to annotate sulfur and nutrient cycling genes and pathways within the ecosystem. This study provides the first comprehensive characterization of the dominant genera in the pool and their metabolic functions, offering new theoretical insights into the microbial processes that regulate sulfur and nutrient dynamics. Understanding the structure and functional roles of microbial communities enhances our knowledge of sulfur cycling and biogeochemical processes in sulfur-rich environments.

## 2. Materials and Methods

### 2.1. Study Area and Sampling Protocol

The Isinuka Spring (31°36′29″ S, 29°28′53″ E), located in Port St. Johns, Eastern Cape Province, South Africa (Figure 1a), is a unique geothermal system previously described by Ncube et al. [6,7]. This study specifically focused on water and sediment samples from the sulfur pool, a bathing pond (Figure 1b) enriched in sulfur compounds. The sulfur pool, measuring approximately 3 m × 3 m and knee-deep, is distinguished by its striking purple-hued water and black, clay-rich sediments. Positioned atop a mountain and encircled by dense vegetation, the sulfur pool remains exposed to direct sunlight. This feature is a perennial water body without any surface tributaries, as its water emerges through seepage from underlying rock formations at the pond’s base. Traditionally, both the water and sediments have been utilized for their reported pelotherapeutic and balneotherapeutic benefits [6,7].

The Isinuka hydrothermal system is influenced by hydrogen sulfide (H_2_S) emissions originating from deep subterranean sources. These emissions mix with groundwater within the upper crust, generating sulfide-rich conditions that sustain the sulfur pool. The continuous release of H_2_S is evident through bubbling activity within the pond, and additional gas fissures emitting H_2_S are present approximately 20 m from the pool, contributing to the site’s characteristic sulfurous odor [6,7]. The sulfur pool, situated atop a mountain and surrounded by dense vegetation, but exposed to sunlight, is a perennial with no tributaries, as its water seeps from underlying rock formations at the pond’s base. Additionally, a small rock cavity, located approximately 20 ft from the sulfur pool, contains clear water that emerges from a rock at its base. This water source, in contrast to the sulfur pool, is free of visible sulfur deposits and purple sulfur bacteria biofilm.

The sampling design involved water sample (n = 3 biological replicates per site) collection (500 mL) from a depth of 5 cm using sterile polypropylene bottles at standardized distances of 150 cm from the sulfur spring’s edge, with sampling points spaced 1 m apart along a transect parallel to the spring perimeter. Sediment samples (n = 3 biological replicates per site; ~300 g) were similarly collected from a depth of 10 cm using a sterile corer, with collection points positioned 50 cm inward from each corresponding water sampling location. For microbial community analysis, DNA was extracted from independently collected replicate samples (n = 3 per matrix). Sediment (~30 g) and water samples (~50 mL) were aseptically collected using sterile disposable spoons or spatulas (pre-packaged, single-use) and immediately transferred to sterile 100 mL screw-capped tubes containing a nucleic acid stabilizing buffer (100 mM EDTA, 100 mM Tris-HCl, 150 mM NaCl, pH 8.0). Field blanks (n = 2 per sampling) consisting of sterile buffer were processed identically to monitor for contamination. All samples were transported on ice (<4 °C) in darkened containers to prevent photodegradation and stored at −80 °C until processing to minimize nucleic acid degradation.

### 2.2. Physicochemical Property Analyses

Physicochemical and elemental analyses were conducted following the methods outlined by Ncube et al. [7], with all measurements performed in triplicate (technical replicates) for each biological sample. In-field measurements of key water quality parameters, including pH, electrical conductivity, salinity, dissolved oxygen (DO), and total dissolved solids (TDS), were completed using a Bante900P multi-parameter water quality meter (Bante Instruments Ltd., Shanghai, China). Laboratory analyses quantified N and S species using standardized colorimetric and titrimetric methods according to the Standard Methods for the Examination of Water and Wastewater [13]. NH_4_^+^ concentrations were determined using the phenate method (APHA 4500-NH_3_ G) with absorbance measured at 640 nm, while NO_3_^−^ and NO_2_^−^ concentrations were quantified using cadmium reduction (Griess assay) (APHA 4500-NO_3_^−^ E) and the direct Griess reaction (APHA 4500-NO_2_^−^ B), with absorbance recorded at 543 nm and 540 nm, respectively. SO_4_^2−^, SO_3_^2−^, and H_2_S were quantified via the turbidimetric method (APHA 4500-SO_4_^2−^ E), iodometric titration (APHA 4500-SO_3_^2−^ B), and the methylene blue method (APHA 4500-S^2−^ D).

Trace elements (including transition/heavy metals), metalloids (As), and light metals (Al, Mg) in water and sediment samples were analyzed using inductively coupled plasma mass spectrometry (ICP-MS 7700 Series) and inductively coupled plasma optical emission spectrometry (ICP-OES 700 Series) (Agilent Technologies, Santa Clara, CA, USA), as previously described by Ncube et al. [7]. Sediment digestion was performed by treating 0.5 g of each sample with 6 mL of nitric acid (HNO_3_) and 1 mL of hydrogen peroxide (H_2_O_2_), followed by microwave-assisted digestion using a MARS microwave digester (CEM Corporation, Matthews, NC, USA). Certified reference materials (NIST 1643e for water, NIST 2711a for sediment) were used to validate recoveries, which were between 90 and 110%.

Key physicochemical parameters, particularly those characterizing sulfur biogeochemistry, are summarized in Table 1.

### 2.3. DNA Extraction, Sequencing, and Bioinformatic Processing

DNA was extracted in triplicate from 0.5 g of sediment and from pelleted water samples (50 mL, centrifuged at 10,000 rpm for 15 min) using the Faecal/Soil Total DNA™ kit (Zymo Research, CA, USA) according to the manufacturer’s protocol. The DNA extract was pooled and used for the amplification of the full-length variable region of the 16S rRNA gene using primers 27F and 1492R under PCR conditions: an initial denaturation at 95 °C for 5 min, followed by 32 cycles of denaturation (95 °C for 1 min), annealing (55 °C for 1 min), and extension (72 °C for 1 min), with a final extension at 72 °C for 7 min [14]. A subsequent nested PCR was performed using primers 515F (5′-GTGBCAGCMGCCGCGGCGGTAA-3′) [15] and 806R (5′-GGACTACNVGGGGTMTCTAATCC-3′) [16], incorporating Illumina-compatible adapters, to amplify the V4 hypervariable region of the 16S rRNA gene [17]. The DNA concentration was quantified with the Qubit dsDNA HS Assay (Life Technologies) so libraries could be pooled to equal molar concentrations for Illumina MiSeq sequencing.

Library preparation and sequencing were conducted on the Illumina MiSeq platform (Illumina Inc., San Diego, CA, USA) using paired-end (300 bp) reads with v3 chemistry, following standard protocols. The raw FASTQ files have been submitted to NCBI’s Sequence Read Archive (SRA) under BioProject ID PRJNA1190128. Amplicon sequence variant (ASV) processing was carried out using the DADA2 pipeline in Nephele (v2.2.8) [18]. During preprocessing, the first 17 bases were trimmed to eliminate residual primers, the truncQ parameter was adjusted to 2, and truncLen was omitted. Chimera filtering was refined by setting the minFoldParentOverAbundance threshold to 8. The taxonomic classification of ASVs was performed using the SILVA REF99 SSU reference database (version 138.1) [19], and the ASV table was converted into a BIOM file using the biom-convert command. Sequences identified as chloroplast or mitochondria sequences were then filtered from the ASV tables. The ASV counts were rarefied to 28,694 reads, the lowest read count, before being used for downstream analyses.

Alpha diversity (Shannon index), evenness (Simpson index), and richness (Chao1 and ACE) analyses were conducted using the R package Phyloseq (v1.24.0) [20] on log-normalized data. Dominant ASVs at the genus level were used to generate a heatmap with the *ampvis2* package [21] in R software (v3.6.1) to visualize the variation and distribution of bacterial communities in the collected samples. Venn diagram analysis, conducted using the *amp_venn* function, was employed to identify shared and unique ASVs within the core microbiome of water and sediment samples. The core microbiome was defined as ASVs present in at least 50% of samples in each group, with a minimum relative abundance of 1% [22].

### 2.4. PICRUSt2 Predictive Function Profiling

The metabolic potential of prokaryotic communities in the sediments and water column of the Isinuka sulfur pool was inferred using the PICRUSt2 algorithm [23], which utilizes 16S rRNA sequence data in conjunction with reference databases such as COG (Cluster of Orthologous Genes) and Kyoto Encyclopedia of Genes and Genomes (KEGG). The analysis was conducted within the QIIME2 framework [24], employing biom-format tables as input for the PICRUSt2 pipeline [23]. Default parameters were applied, including EPA-NG for sequence placement within a reference phylogenetic tree, maximum parsimony for hidden-state prediction, and an NSTI (Nearest-Sequenced Taxon Index) cut-off of 2.0.

PICRUSt2 generates a predictive metagenome by integrating ASV taxonomic classifications with reference genome data, enabling functional inferences about the microbial community [25]. Functional annotations were assigned based on the KEGG [26] and the MetaCyc pathway database [27], with metabolic pathways resolved to Level 3 Orthology. Additionally, genes associated with carbon, nitrogen, and sulfur cycling were predicted. The results were visualized using bar plots and heatmaps generated with the *ggplot2* and *ampvis2* R packages.

## 3. Results and Discussion

### 3.1. Physicochemical Parameters of the Sediment and Water

The sulfur biogeochemistry of the Isinuka sulfur pool revealed distinct spatial variations between the sediment and water column, driven by differences in physicochemical conditions such as salinity, dissolved oxygen (DO), total sulfur (TS), sulfur species (SO_3_^2−^ and H_2_S), and heavy metal composition (Table 1).

Sediments exhibited a slightly higher pH (8.6) compared to the water column (8.1), along with significantly elevated TS (3470 mg/kg vs. 1958 mg/L, *p* < 0.001) and TDS (3424 mg/kg vs. 1879 mg/L, *p* < 0.001), indicative of a more concentrated sulfur-rich environment. The elevated salinity in sediments (5.0 g/kg vs. 3.0 g/kg in water, *p* < 0.001) suggests prolonged S accumulation, likely influenced by mineral precipitation and microbial SO_3_^2−^ reduction processes. Both the sediments and water column were largely anoxic, with no detectable DO in the sediments, while the water column exhibited only a minimal DO level (0.27 ± 0.107 mg/L). Given the high enrichment of purple sulfur bacteria (PSB) in the water column, it is likely that the detected DO resulted from atmospheric contamination during sampling. Otherwise, bleaching of the PSB would have been observed due to the sensitivity of the photosynthetic pigments to aerobic conditions [28].

Similarly, SO_4_^2−^ and SO_3_^2−^ levels were significantly higher in sediments (139 mg/kg and 1.72 mg/kg, respectively) compared to the water column (77 mg/L and 0.34 mg/L). This pattern suggests the possibility of active SO_4_^2−^ reduction in the sediments, potentially facilitated by sulfate-reducing bacteria (SRB) that convert SO_4_^2−^ into S^2−^ under anoxic conditions. However, H_2_S concentrations displayed an inverse distribution, being markedly higher in the water column (15.3 mg/L) than in sediments (4.98 mg/kg). This suggests that while sulfate reduction occurs predominantly in sediments, upward diffusion of H_2_S into the water column, coupled with possible chemotrophic or phototrophic S oxidation, maintains elevated H_2_S levels in the water. These findings suggest that the Isinuka sulfur pool exhibits a dynamic sulfur cycle, with sediments functioning as a primary site for SO_4_^2−^ reduction and S mineralization, while the water column supports S^2−^ oxidation. The interplay between these microbial processes contributes to the distinct sulfur chemistry observed across the two habitats. These observations align with prior reports on the Isinuka sulfur spring [5,6,7] and analogous sulfur-rich environments globally [11,29,30,31].

Nitrogen species (NH_4_^+^, NO_3_^−^) and major anions (Cl^−^, SO_4_^2−^, SO_3_^2−^) were predominantly concentrated in the sediments, reflecting active sulfur-mediated redox cycling and nutrient sequestration. Similarly, heavy metal partitioning revealed distinct geochemical behaviors: metals such as Fe, Mn, and Zn were significantly enriched in sediments, consistent with their precipitation under reducing conditions and their affinity for particulate phases. In contrast, higher concentrations of As and Mg in the water column suggest enhanced solubility and potential mobilization in sulfur-rich environments. Notably, the extreme Fe concentrations in sediments (>24,000 mg/kg) emphasize its role as a critical geochemical sink. However, the elevated arsenic levels in the water (>3-fold higher than in sediments) raise concerns about its bioavailability and potential toxicity, especially given that local communities use this water for its reputed healing properties. These findings corroborate prior assessments by Ncube et al. [7] and Faniran et al. [5], who documented elevated heavy metal concentrations in the spring water.

### 3.2. Diversity of Bacterial Communities

After quality control and chimera removal, a total of 77,348 high-quality 16S rRNA sequences was obtained, representing 291 ASVs in water samples and 1094 ASVs in sediment samples (Table 2). Good’s coverage values ranged from 99.34% to 99.90%, indicating that the sequencing depth was sufficient to capture the majority of microbial diversity. Rarefaction curve plots approached asymptotic plateaus (Figure 2a), further validating the comprehensive characterization of bacterial communities. Notably, 41.9% of sediment reads could not be classified at the genus level, compared to just 3.93% in water samples. This indicates a higher prevalence of novel or poorly characterized taxa in sediments, highlighting the potential for undiscovered functional microbial groups that warrant further investigation for their ecological roles and applications in biogeochemical cycling.

Comparative analysis of alpha diversity indices further revealed significantly higher microbial richness and diversity in sediment samples than in water samples. This was reflected in the higher ACE, Chao1 (richness), Shannon (diversity), and Simpson (evenness) values for sediments (Table 2). These findings align with previous studies, including those by Cole et al. [32], who reported higher microbial diversity in the sediments than in the water column of sulfur-rich hot springs. The higher diversity in sediments can be attributed to several localized ecological factors. Sediments provide a heterogeneous microenvironment characterized by physicochemical gradients, including variations in oxygen availability, organic matter accumulation, and sulfur compound sequestration. These factors create specialized microbial niches that support greater taxonomic and functional diversity. The presence of redox gradients, ranging from microaerophilic surface layers to deeper anoxic zones, enables the coexistence of aerobic, facultative anaerobic, and obligate anaerobic microbes [1,33].

Furthermore, sediments serve as reservoirs for organic matter, promoting heterotrophic and chemolithotrophic microbial activities [4]. These processes often involve syntrophic interactions and cross-feeding among microbes. Sulfur compounds such as S°, SO_3_^2−^, and SO_4_^2−^, which are abundant in sulfur springs, are preferentially sequestered in sediments, promoting the growth of sulfur-metabolizing microbes like sulfur-oxidizing bacteria (SOB) and sulfate-reducing bacteria (SRB). These processes contribute to broader biogeochemical cycles, influencing nutrient availability and ecosystem productivity. In contrast, the water column exhibits dynamic physicochemical conditions, including fluctuating dissolved oxygen (DO) levels and increased hydrodynamic mixing, which can constrain microbial colonization and stability [24,25]. The dominance of r-strategists—microbes adapted to rapid growth under fluctuating conditions—in water samples further supports this observation, whereas sediment communities contain both r-strategists and K-strategists, which thrive in more stable environments [34].

The enrichment of novel taxa in sediments underscores the importance of sulfur springs as reservoirs of microbial biodiversity, potentially harboring taxa with unique metabolic pathways relevant to biogeochemical cycling and biotechnological applications. Microbial communities in these systems likely play crucial roles in sulfur detoxification, metal bioremediation, and organic matter decomposition, which could be exploited for environmental management strategies.

### 3.3. Taxonomic Composition of Bacterial Community

The bacterial taxa identified in the Isinuka sulfur pool microbiome spanned 54 phyla, 114 classes, 226 orders, and 920 genera, underscoring the taxonomic complexity of these microbial communities. However, the distribution of these taxa varied significantly between the water and sediment samples. The analysis of relative abundances revealed that water samples harbored 19 bacterial phyla, dominated overwhelmingly by Proteobacteria (95.5%), followed by Actinobacteria (1.66%), Planctomycetes (1.16%), and Firmicutes (1.12%). Fifteen additional phyla were detected at relative abundances below 0.5%, indicating a highly skewed community structure (Figure 2b). Within Proteobacteria, members of class γ-Proteobacteria (67.29%), α-Proteobacteria (24.49%), and β-Proteobacteria constituted the dominant bacteria taxa in the water column (Figure 2c). In contrast, sediment samples exhibited a far more diverse microbial community, comprising 55 phyla. While Proteobacteria remained the most abundant phylum, its relative dominance was reduced to 66.04%, with other phyla contributing significantly to community composition. These included Firmicutes (10.7%), Bacteroidetes (6.4%), Actinobacteria (2.4%), Planctomycetes (1.6%), Chloroflexi (1.5%), Tenericutes (1.42%), TM6 (1.3%), and SR1 (1.19%). Additionally, four other phyla were detected at relative abundances exceeding 0.5%, underscoring the greater taxonomic complexity of sediment-associated microbial communities (Figure 2b).

At the class level, sediments displayed greater microbial diversity within the aforementioned phyla (Figure 2c). For example, all recognized classes of Proteobacteria (α-, β-, δ-, γ-, ζ-, and ε-Proteobacteria) were detected across the two niches, with relative abundances ranging from 0.025% to 27.3%. However, members of Bacteroidetes, including the classes Sphingobacteriia, Flavobacteria, Cytophagia, and Bacteroidia, were more prevalent in sediments, whereas only members of Cytophagia were found in water samples. Interestingly, similar members of Firmicutes were detected in both water and sediment samples, though sediment samples exhibited markedly higher species richness within this phylum.

Figure 3 and Appendix A illustrate the genus-level distribution of bacterial taxa in the water and sediment samples, revealing distinct microbial community structures. The phylogenetic relationship between the main genera is also presented in Figure 3b. The water column was dominated by *Thiocapsa* (65.57%), a key sulfur oxidizer and anoxygenic photoautotroph [35,36]. Other significant taxa in the water column included *Delftia* (2.23%), involved in sulfur and organic compound degradation [37]; *Hoeflea* (1.00%), a nitrogen-fixing oil-degrading anoxygenic photoautotroph [38]; and *Legionella* (0.11%), associated with biofilm formation in aquatic habitats [39,40]. In contrast, the sediments supported a more diverse and evenly distributed community. *Delftia* (8.27%) was significantly more abundant in sediment, reflecting its adaptation to organic-matter-rich conditions. *Thiocapsa* was less dominant in the sediment (2.44%), consistent with its preference for oxygenated environments, despite undertaking anoxygenic photosynthesis using H_2_S or S^o^ as the electron donor [35,36]. Sediments also contained higher levels of sulfidogenic taxa, such as unclassified Coxiellaceae (8.03%), unclassified *Thioprofundaceae* (2.78%), unclassified Desulfobacteraceae (2.43%), and *Tissierella* (1.41%) [1,4,41], along with heterotrophic and facultative taxa like *Idiomarina* (2.21%), which likely contribute to organic carbon cycling in sediment [42,43]. Sediments, with resource-rich, stable microhabitats, support a diverse microbial community [2,4,33], while the nutrient-limited water column favors a less diverse, Proteobacteria-dominated assemblage (Figure 3). These patterns highlight the influence of environmental heterogeneity on microbial structure and function.

The core microbiome analysis, visualized through a Venn diagram (Figure 3a, inset), revealed 19 ASVs shared between the water and sediment samples, constituting 60.9% of the total valid sequences. These core taxa, defined as ASVs at the genus level present in at least 50% of samples in each group with a minimum relative abundance of 1%, highlight the functional backbone of the sulfur pool’s microbial ecosystem. Key members of the shared core microbiome included *Thiocapsa*, *Delftia* (*D. litopenaei*), and *Lutimaribacter* (*L. litoralis*). The dominance of *Thiocapsa* (primarily *T. marina* and *T. litoralis*, along with unclassified *Thiocapsa* species) in the water column, where it constitutes 65.57% of the SMB community, underscores its role as a keystone taxon. *Thiocapsa* facilitates sulfur oxidation and drives biogeochemical cycling under dynamic, oxygenated conditions. Notably, *T. marina* and *T. litoralis*, previously isolated from brackish-to-marine sediments of the Mediterranean Sea, the White Sea, and the Black Sea, are alkaliphilic, purple-red, okenone-containing anoxygenic photoautotrophs [35,36]. These species utilize sulfur compounds as electron donors during photosynthesis and have the capacity to oxidize sulfide, thiosulfate, and other reduced sulfur compounds [1,4]. This study represents the first report of *Thiocapsa* as a keystone taxon in a sulfur-enriched spring environment. In addition, *Lutimaribacter* species, known for their capability to degrade cyclohexylacetate, have previously been isolated from seawater [44]. Recently, Somee et al. [45] reported the enrichment of *Lutimaribacter* in sulfur-rich oil spills in the marine ecosystem of the Persian Gulf.

Other minor shared core microbiomes included *Sneathiella* (*S. chungangensis*), *Sphingorhabdus* (*S. marina*), *Hoeflea* (*H. alexandrii*, and *H. halophila*), *Bradyrhizobium* (*B. japonicum*), and unclassified *Halochromatium*, among others (Appendix A). Similar to the genus *Thiocapsa*, *Halochromatium* facilitate the transformation of reduced sulfur compounds into oxidized forms critical for maintaining sulfur equilibrium in the ecosystem [46,47] by linking aerobic and anaerobic sulfur cycling across niches. Organic compound degraders like *Sneathiella* contribute to carbon mineralization and organic matter turnover, enhancing nutrient recycling and energy flow [48]. In contrast, nitrogen-related taxa such as *Bradyrhizobium* and *Hoeflea* link sulfur and nitrogen cycles through processes like nitrogen fixation and nitrate reduction, sustaining primary production and microbial growth in nutrient-limited conditions [4]. The presence of *Hoeflea* in both water and sediment links sulfur, iron, nitrogen, and carbon cycles in the Isinuka sulfur pool. Members of this genus have been reported to be part of an iron-oxidizing, mixotrophic taxon capable of anaerobic nitrate and nitrous oxide reduction [49], contributing to denitrification, carbon mineralization, and iron–sulfur cycling. Thus, this taxon promotes nutrient recycling and the coupling of sulfur and nitrogen transformations, supporting ecosystem stability.

The study also identified unique niche-specific core microbiomes in the sediment and water samples (Figure 3a Inset; Appendix A). The sediment supported 136 unique core microbiomes, comprising 24.8% of valid reads—predominantly sulfur-reducing bacteria and organic/inorganic degraders. These taxa highlight the sediment’s specialization for anoxic, resource-rich environments, playing pivotal roles in sulfur cycling, organic matter decomposition, and ecosystem stability. The dominance of anoxic sulfur-reducing genera aligns with the low oxygen availability in sedimentary habitat, highlighting its role as a hotspot for anaerobic biogeochemical processes. In contrast, water samples exhibited only 10 unique core genera, including *Roseovarius* (*R. mucosus*), *Gimesia* (*G. maris*), unclassified *Pusillimonas*, *Sulfitobacter* (*S. mediterraneus*), and unclassified *Thiocapsa*, accounting for 2.1% of total sequence reads. These taxa demonstrated adaptations to dynamic conditions, with *Sulfitobacter* and *Thiohalocapsa* specializing in sulfur oxidation, critical for sulfur transformations in the water column [12,36,46], and *Roseovarius* and *Gimesia* contributing to organic sulfur compound degradation [50], maintaining ecological balance in this niche. The smaller number of unique core microbiomes in water reflects its selective and competitive environment, favoring specialized taxa suited for nutrient-limited, dynamic conditions.

### 3.4. Stratification of Sulfur-Oxidizing (SOB) and Sulfate-Reducing Bacteria (SRB)

The spatial distribution and diversity of sulfur-metabolizing bacteria (SMB) further underscore these niche specializations (Figure 4). SMB communities exhibited marked differentiation between water and sediment samples, shaped by environmental gradients. These bacteria were categorized into SOB and SRB, with SOB further subdivided into green sulfur bacteria (GSB), purple sulfur bacteria (PSB), and other SOB taxa. This stratification reflected the distinct ecological roles and adaptations of SMB to the sulfur pool’s dynamic conditions.

As illustrated in Figure 4a, GSB were exclusively found in sediment samples, thriving in anaerobic, sulfur-rich microhabitats. Key taxa included *Chlorobaculum* (0.115%), unclassified FJ517031 (0.310%), and unclassified EF471569 (0.187%), with trace detections of *Melioribacter* and unclassified FN429794 (<0.02%). These bacteria likely perform anoxygenic photosynthesis under low-light, anoxic conditions characteristic of deeper sediment layers [51,52]. In contrast, PSB were distributed across both niches, with *Thiocapsa* dominating the water column (65.57%) but present in sediments at lower levels (2.44%) (Figure 3a). Other PSB, such as unclassified Chromatiaceae and *Halochromatium*, were detected in both environments but at reduced abundances. Habitat-specific taxa like *Thiohalocapsa* (exclusive to water) and *Ectothiorhodosinus* (sediment-specific) reflected adaptations to light and oxygen gradients. Beyond PSB, other SOB taxa displayed niche-specific distributions (Figure 3a and Figure 4a). For example, *Delftia* was more abundant in sediments (8.27%), while *Lutimaribacter* dominated the water column (17.87%) but was scarce in sediments (0.13%). Sediment-specific genera, including unclassified *Sulfurimonas* (1.53%), unclassified *Thioprofundaceae* (2.78%), and *Thioalkalivibrio* (0.40%), were found, highlighting their adaptation to sulfur-rich, anoxic conditions and their contributions to sulfur oxidation and cycling [53,54]. In contrast to SOB, SRB were exclusively confined to sediments, reflecting their strict anaerobic requirements. Dominant taxa included unclassified Desulfobacteraceae (2.43%), *Desulfosarcina* (0.25%), and *Desulfococcus* (0.27%), supported by minor contributors like *Dethiosulfatibacter* (0.16%) and *Desulfatiglans* (0.47%). These SRB play pivotal roles in sulfate and sulfur reduction, facilitating biogeochemical cycling by providing substrates (e.g., hydrogen sulfide) that support SOB metabolism and fostering syntrophic interactions between these groups [1,2,12,55].

The partitioning of SMB reflects the adaptation of these taxa to their respective ecological niches. The water column, characterized by oxygen-rich and dynamic conditions, supported phototrophic SOB like *Thiocapsa* and *Lutimaribacter*. Sediments, with their stable, anoxic environments and sulfur-rich conditions, fostered diverse SMB communities, including anaerobic SOB and SRB, enabling a tightly coupled sulfur cycle [4,12]. This partitioning reflects the adaptation of SMB to their respective ecological niches, with sediment-associated taxa playing critical roles in sustaining a tightly coupled sulfur cycle through syntrophic interactions [1,2,12,55]. Microbial diversity and core microbiome analyses highlight functional linkages between water and sediments, facilitating cross-niche processes such as sulfur and nitrogen cycling. While the shared core microbiome reflects these interconnections, the unique microbiomes of each niche demonstrate adaptations to distinct environmental factors like oxygen nutrient gradients and habitat stability. Sediment-associated taxa, enriched in sulfur-reducing and organic-matter-degrading bacteria, contribute to ecosystem stability under resource-rich and anoxic conditions, serving as hotspots for anaerobic biogeochemical processes. Meanwhile, water column taxa, specialized for dynamic anoxic/oxic conditions, emphasize sulfur oxidation and organic matter degradation. This complementarity between niches maintains the biogeochemical balance of the sulfur pool, with broader implications for ecosystem resilience and sustainability.

### 3.5. Potential Functional Diversity and Distribution Patterns

PICRUSt2 utilizes 16S rRNA gene sequencing data to infer potential microbial functions (38), and has been widely applied in analyzing microbial functional profiles in sulfur-rich ecosystems [56,57], as well as various other environments [58,59,60]. In this study, PICRUSt2 was employed to predict the metabolic capabilities of prokaryotic communities within the sediments and water column of the Isinuka sulfur pool (Figure 5). On top of this, diverse genes involved in the C, N, and S cycles were also predicted (Appendix A).

The analysis revealed 346 MetaCyc pathways (level 3) distributed across 40 level 2 superclasses and 7 level 1 KEGG groups (Appendix A). In both sediment and water samples, 15 superclasses contributed over 90% of the total relative abundance. Dominant pathways included amino acid (20.7%) and nucleotide biosynthesis (14.8%), followed by lipid (10.4%) and vitamin biosynthesis (8.85%), energy metabolism (8.4%), degradation (5.1%), carbohydrate biosynthesis (4.3%), and cell structure biosynthesis (3.5%). Additional pathways encompassed fermentation (3.4%), cofactor biosynthesis (2.7%), and secondary metabolite production (2.3%), along with amino acid degradation (2.2%), aromatic compound breakdown, sulfur metabolism (1.3%), and polyamine biosynthesis (1.1%). Notably, the relative abundance of cofactor biosynthesis pathways was significantly higher in the water column (5.34%) compared to sediments (0.72%). This disparity suggests that microbial communities in the water column are adapted to dynamic and nutrient-limited conditions, necessitating enhanced cofactor production to support the critical enzymatic activities involved in sulfur metabolism and energy production. This metabolic adaptation may also reflect a strategy to optimize microbial functionality in the highly competitive and stressful anoxic aquatic environment of the sulfur spring [61].

### 3.6. Biogeochemical Cycling

#### 3.6.1. Carbon and Methane Metabolism

Elemental metabolism revealed that genes involved in carbon fixation pathways were more abundant in the water column compared to sediments (Figure 5a; Appendix A). Among these, the *accA* gene, encoding acetyl-coenzyme A (CoA) carboxylase—a key enzyme in the 3-hydroxypropionate/4-hydroxybutyrate (3-HP/4-HB) pathway for CO_2_ fixation—was prevalent in both niches (Figure 5b). This pathway is particularly significant for autotrophic archaea and extremophilic bacteria that fix CO_2_ to synthesize essential cellular components. Acetyl-CoA carboxylase catalyzes the conversion of acetyl-CoA to malonyl-CoA, a critical early step in the 3-HP/4-HB pathway. The abundant presence of *accA* in both niches underscores the importance of the 3-HP/4-HB pathway for CO_2_ fixation, primarily attributed to autotrophic or mixotrophic organisms such as members of the phylum Chloroflexi (e.g., Anaerolinaceae, Thermomicrobia, and Caldilineaceae) and β-Proteobacteria within the order Burkholderiales (e.g., *Alicycliphilus*, *Lautropia*, unclassified Comamonadaceae, and *Parapusillimonas*) [62]. In extreme environments like sulfur springs and hydrothermal vents, archaeal groups such as Thaumarchaeota and some Crenarchaeota, though not characterized in this study, likely utilize the 3-HP/4-HB pathway for autotrophic carbon fixation [63]. Interestingly, the *abfD* gene encoding 4-hydroxybutyryl-CoA dehydratase was absent in both niches. This enzyme is critical for converting 4-hydroxybutyryl-CoA into crotonyl-CoA, an essential intermediate in the 3-HP/4-HB pathway [55,62]. The absence of *abfD* suggests a potential metabolic gap in the complete pathway. However, it is plausible that uncharacterized enzymes or analogous pathways compensate for this missing step. Another possibility is that organisms in these niches rely on partial utilization of the 3-HP/4-HB pathway, integrating steps from other CO_2_ fixation pathways such as the reductive acetyl-CoA pathway. Supporting this hypothesis, three key genes of the reductive tricarboxylic acid (rTCA) cycle—*aclB*, *acsAB*, *pccAB*, *oorA*, and *por/nifJ*—were also predicted to be more abundant in both the water column and sediments. This observation aligns with previous reports that microorganisms utilizing the rTCA cycle for CO_2_ fixation are ubiquitous in microaerophilic to anaerobic, sulfur-rich environments [3,57,63].

As illustrated in Figure 5a, methane metabolism genes were exclusively detected in sediments (0.0038%), indicating that methane-related processes, such as methanogenesis or anaerobic methane oxidation, are confined to the anoxic sediment environment. This finding emphasizes the sediments’ critical role in methane cycling and its potential influence on greenhouse gas dynamics. Furthermore, genes encoding copper-dependent particulate methane/ammonia monooxygenase (pMMO), essential for methane oxidation (*pmoA-amoA*, *pmoB-amoB*, and *pmoC-amoC*), were detected only in sediment samples. These genes are likely contributed by methane-oxidizing bacteria belonging to *α-proteobacteria* and *γ-proteobacteria* in anaerobic, sulfur-rich habitats. These genes are likely attributed to methane-oxidizing bacteria belonging to α- and γ-proteobacteria, which—though classically aerobic—are known to persist in microaerophilic or anoxic environments by employing fermentation-based methanotrophy or facultative anaerobic pathways, such as denitrification [63,64].

#### 3.6.2. Nitrogen Metabolism

Nitrogen cycling in aquatic ecosystems involves critical processes like assimilatory nitrate reduction, denitrification, dissimilatory nitrate reduction (DNRA), nitrification, and nitrogen fixation. The distribution of key functional genes associated with these processes varies significantly between sediments and the water column, reflecting niche-specific adaptations and metabolic dynamics (Figure 5, Appendix A). Nitrogen metabolism genes were more abundant in the water column (0.97%) than in sediments (0.54%), but distinct patterns emerged across processes and environments.

Assimilatory nitrate reduction genes, such as *nasA*, *nirA*, and *narB*, were detected in both environments, with consistently higher abundance in sediments (Figure 5c). For instance, *nirA* (reducing nitrite to ammonium) was threefold more abundant in sediments (0.0334%) than the water column (0.0092%), highlighting the sediment’s nutrient-rich, anaerobic conditions that favor microbial growth. In the water column, *Thiocapsa*, capable of nitrate assimilation and sulfur metabolism [35,46,65], and *Lutimaribacter*, associated with organic nitrogen utilization in marine systems [44], were the key assimilatory nitrate reducers. In the sediments, *Bradyrhizobium* was the main taxa linked to the assimilatory nitrate reduction.

Denitrification genes demonstrated stark contrasts between the two habitats. Genes such as *narG*, *narH*, and *narI*, which encode nitrate reductase subunits, were detected exclusively in sediments (~0.0168%), indicating sediment-specific denitrification activity (Figure 5c). By contrast, genes like *nirS* (0.2928%) and *norC* (0.3004%), which encode nitrite and nitric oxide reductases, respectively, were significantly enriched in the water column. In sulfur-rich anoxic sediment with limited organic matter, microbes utilize mainly sulfur autotrophic denitrification (SADN), a biochemical process where microorganisms use reduced sulfur compounds (e.g., H_2_S, S_2_O_3_^2−^, or S^o^) as electron donors to reduce nitrate (NO_3_^−^) or nitrite (NO_2_^−^) to nitrogen gas (N_2_) [4,33,66]. The key taxa enriched in the sediment samples that have previously been reported to undertake SADN under diverse environments include *Paracoccus*, *Sulfurimonas*, *Pseudorhodobacter*, *Thioalkalivibrio*, *Rhodobacter*, and *Halomonas* [11,12,67,68,69].

Similar to denitrification, DNRA genes showed higher abundance in sediments. Key genes such as *narG*, *napA*, and *napB* were significantly enriched in sediments (~0.0670%), while the water column had minimal contributions (~0.0076%). Notably, genes encoding nitrite reductases (*nirB* and *nirD*) were present in both habitats but slightly more abundant in sediments, respectively. Interestingly, the presence of *nrfA* and *nrfH*, albeit at low levels (0.0072%), was exclusive to the water column, suggesting a potential niche-specific role in nitrite reduction under microaerophilic conditions. Key taxa such as *Sulfurimonas* (0.7173%), *Pseudorhodobacter*, *Rhodobacter*, and *Halomonas* were the main denitrifiers identified that have been reported elsewhere in sulfur-rich ecosystems [63,70]. Nitrification genes, encoding for particulate methane/ammonia monooxygenase (*pmoA-amoA*, *pmoB-amoB*, and *pmoC-amoC*), were detected in both environments but showed slightly higher relative abundance in the water column. This indicates active ammonia oxidation in both niches, with potentially greater microbial activity in the dynamic, oxygen-rich water column. *Nitrosomonas* (0.9516% abundance in sediments) was the key ammonia-oxidizing bacteria identified as driving the first step of nitrification. *Nitrotoga* (0.7009% abundance in sediments), which participates in nitrite oxidation during nitrification, often in low-temperature environments, was the other taxa identified in both habitats. In addition, facultative methylotrophic genera *Methylobacterium* (0.1953%) and *Methyloceanibacter* (0.224%) may contribute to the coupling of carbon and nitrogen cycles through the metabolism of C1 compounds (like methanol, formaldehyde, or sometimes methane derivatives) and participation in nitrogen transformations [71]. Nitrogen fixation genes (*nifA*, *nifB*, *nifD*, *nifE*, *nifH*, *nifK*, and *nifN*) were present in both habitats but displayed a higher overall abundance in sediments (Figure 5c), mainly driven by taxa such as *Bradyrhizobium*, *Hoeflea*, *Clostridium,* and *Mesorhizobium*. This pattern highlights the sediments as a key reservoir for nitrogen-fixing microbes, likely due to the availability of organic matter and the microaerophilic-to-anaerobic conditions favorable for nitrogenase activity. However, the presence of these genes in the water column suggests nitrogen fixation by planktonic microbes in oxygen-poor microenvironments.

Collectively, sediments harbor genes central to assimilatory and dissimilatory nitrate reduction and nitrogen fixation, underscoring their role in nitrogen storage and transformation under anaerobic conditions. The water column, enriched in denitrification (*nirS*, *norC*) and nitrification genes, supports active nitrogen cycling driven by oxygen gradients and dynamic microbial communities. Processes like SADN link sulfur and nitrogen cycles, while organic carbon oxidation often fuels denitrification and DNRA, completing N-C-S coupling [4,70]. This integration maintains nutrient balance and energy flow, emphasizing the interdependence of microbial processes across aquatic ecosystems [12].

#### 3.6.3. Sulfur Metabolism

Sulfur, a highly redox-active element capable of existing in oxidation states from +6 to −2, is essential for various biochemical processes, including energy production and carbon cycling. The sulfur cycle encompasses several key pathways: (1) dissimilatory sulfate reduction (DSR; SO_4_^2−^ to sulfide); (2) assimilatory sulfate reduction (ASR; reducing sulfite to sulfide via *cysI* or *sir* genes); (3) sulfur oxidation (via *fccAB*, *glpE*, or *sqr*); (4) thiosulfate oxidation (via SOX system and *tsdA/doxAD*); and (5) sulfite oxidation (via *soeABC/sorAB*) [1,4,12,55]. In this study, sulfur metabolism genes were slightly more abundant in the water column (1.39%) than in sediments (1.26%) (Figure 5a). Both environments demonstrated a relatively complete repertoire of sulfur-cycle-related pathways, with subtle habitat-specific distinctions. The water column showed an enrichment of genes involved in thiosulfate oxidation via the SOX system, while dissimilatory and assimilatory sulfate reduction pathways were more prominent in sediments (Figure 6a). Interestingly, sulfur oxidation pathways mediated by fccAB and thiosulfate oxidation via *tsdA* were comparable across both niches. However, no evidence of sulfite oxidation pathways mediated by *soeABC/sorAB* was detected in either environment. Nevertheless, this comprehensive repertoire of sulfur cycle pathways highlights the distinct yet overlapping roles of microbial communities in sulfur transformations, including phototrophic processes such as anoxygenic photosynthesis.

By correlating the relative abundances of key genera with their known metabolic capacities (Appendix A), anoxygenic photosynthesis emerges as a pivotal microbial process in the Isinuka sulfur pool ecosystem, coupling light energy to sulfur metabolism and carbon fixation [11,55]. This process, carried out by phototrophic bacteria, uses light energy to drive photosynthesis without producing oxygen. Instead of water, these bacteria typically rely on reduced sulfur compounds (e.g., H_2_S) or organic compounds as electron donors [1,4,12]. The higher abundance of purple sulfur bacterial genera, such as *Thiocapsa*, *Thiohalocapsa*, *Halochromatium*, and unclassified members of the family Chromatiaceae, along with some *α-Proteobacteria* from the family Rhodobacteraceae (e.g., *Pseudorhodobacter*, *Roseovarius*), strongly supports the occurrence of anoxygenic photosynthesis in the water column of the Isinuka sulfur pool. These taxa were linked to a suite of genes encoding proteins and enzymes that facilitate the absorption of light (*pufLM*, *pucAB*, and *crt*), electron transport (*cycA*, *petA*, *petB*, and *petC*), regulation (*pprAB* and *fnr*), and the synthesis of energy-rich compounds like ATP and NADPH (*cbbL*, *cbbS*, *cbbP*, and *cbbA*), and utilize reduced sulfur compounds (H_2_S, S^0^, or S_2_O_3_^2−^) as electron donors (soxABCD) (Figure 6, Appendix A). In contrast, anoxygenic photosynthesis in sediments was supported mainly by green sulfur bacteria, including *Chlorobaculum*, unclassified taxa (*FJ517031*, *EF471569*), *Melioribacter*, and unclassified FN429794. These organisms likely utilize reduced sulfur compounds, such as sulfide, elemental sulfur, or thiosulfate, as electron donors under low-light anoxic sulfide conditions using type I reaction center light harvesting (*psaA/B/C/D/E*, *bch*, and *crt*), cytochrome bc1 complex (*petA/B/C/D*) ferredoxin and cytochrome proteins (*fdx* and *cyd*), and reductive rTCA cycle (*aclA/B* and *sdh*) and regulatory and accessory genes (*fmoA* and *cbb*). This process is crucial for sustaining sulfur cycling and energy flow in this unique ecosystem, emphasizing the ecological significance of these microbial communities.

Extending these observations, a detailed analysis of sulfur oxidation pathways reveals the genetic underpinnings of sulfur cycling, with a focus on genes such as *soxABCXYZ* and *sqr*, which drive thiosulfate and sulfide oxidation, respectively (Figure 6). The *sqr* gene encodes sulfide quinone oxidoreductase, a membrane-bound protein that plays a crucial role in energy production for sulfur-oxidizing bacteria (SOB). Variants such as *sqrF* and *sqrD* are commonly found in purple sulfur bacteria (PSB) and green sulfur bacteria (GSB) [55,65]. Similarly, the *soxABCXYZ* gene cluster, encoding proteins crucial for thiosulfate (S_2_O_3_^2−^) oxidation [4,52,55], is prevalent in various PSB across diverse ecosystems [2,52,65]. In circumneutral sulfur springs (pH 8.0–8.5), S_2_O_3_^2−^ availability plays a critical role in shaping dominant SOX pathways and associated geochemical outcomes [72]. These pathways are frequently associated with α-, β-, and γ-Proteobacteria, including *Chlorobi* SOB, under oxygenated conditions [2,4,55,72]. In this study, *γ-Proteobacteria*, such as *Thiocapsa* and *Thiohalocapsa*, unclassified Chromatiaceae, and *Halochromatium*, dominated the water column (Figure 6b). These taxa significantly contribute to the *soxABCXYZ* cluster and *sqr*, driving thiosulfate oxidation and sulfur transformations. This aligns with the findings by Petukhova et al. [65] that the genus *Thiocapsa* (*T. bogorovii*) possesses multiple homologs of the *soxABCXYZ* cluster and *sqr*, forming an elaborate SOX system and supporting sulfide quinone oxidoreductase (SQR) activity without sulfur accumulation. Furthermore, their study highlighted the presence of multiple flavocytochrome c sulfide dehydrogenase (*fccAB*) homologs, which complement the SOX system and SQR processes. While *fccAB* abundance was similar in water and sediment samples (Figure 6a), sediment samples harbored *Chlorobaculum*, unclassified FJ517031, unclassified EF471569, *Melioribacter*, and unclassified FN429794, which contributes to *fccAB* activity (Figure 6b). These taxa are key contributors to SOX and SQR activities under the low-light, anoxic conditions characteristic of deeper sediment layers [12,51,52].

Building on the microbial processes driving sulfur oxidation, we next examined the genetic basis of sulfur assimilation through the assimilatory sulfate reduction (ASR) pathway, a critical step in microbial sulfur metabolism. Assimilatory sulfate reduction (ASR), the primary mechanism enabling microbes to incorporate environmental sulfur into cellular components, plays a pivotal role in microbial sulfur metabolism [4,47]. ASR is the primary source of cysteine biosynthesis, a critical pathway for heavy metal detoxification in microbial systems. Two main strategies are employed in the ASR pathway: (1) adenosine-5’-phosphosulfate (APS) is phosphorylated by adenylylsulfate kinase (*CysC*) to form 3’-phosphoadenosine-5’-phosphosulfate (PAPS), which is subsequently reduced to sulfite (SO_3_^2−^) by PAPS reductase (*CysH*); or (2) APS is directly reduced by APS reductase (*aprAB*) to yield adenosine monophosphate (AMP) and SO_3_^2−^ [1,2]. In both cases, sulfite is further reduced by either anaerobic sulfite reductase (*asrABC*) or sulfite reductase (*CysJI*) to produce sulfide (S^2−^), which is incorporated into L-cysteine via cysteine synthase A (CysK) and cysteine synthase B (CysM) [4]. In this study, sediments exhibited higher levels of genes such as sulfate adenylyltransferase and adenylylsulfate kinase (*cysNC*), sulfate adenylyltransferase subunit 2 (*cysD*), PAPS reductase (cysH), and sulfite reductase (*cysI*) (Figure 6a,b), suggesting a diverse and robust ASR system. This enrichment likely reflects the elevated organic matter content and anaerobic conditions in sediments, which demand efficient sulfur assimilation. In contrast, the water column showed an enrichment of *cysH* and *cysI*, as well as *aprAB* genes, indicating a streamlined sulfur assimilation process optimized for dynamic, oxygen-rich conditions. This pathway was predominantly associated with taxa belonging to *Clostridium*, *Bacillus*, *Beijerinckia*, and *Leucobacter*, highlighting niche-specific adaptations in ASR dynamics.

While the ASR pathway facilitates sulfur incorporation into cellular components, dissimilatory sulfite metabolism (DSR), powered by sulfite reductases (Dsr), enables energy generation through the reduction of sulfite to sulfide, playing a central role in sulfur cycling across diverse ecosystems [12]. Dissimilatory sulfite reduction (DSR) serves as a key intracellular catabolic mechanism employed by sulfate-reducing bacteria (SRB). Unlike ASR, which incorporates sulfate into sulfur-containing amino acids, DSR uses sulfite as an electron acceptor to generate energy, reducing it to sulfide in the process. This pathway enables SRB to utilize organic compounds as both carbon and energy sources, establishing a vital link between sulfur cycling and broader carbon and energy dynamics [4,12]. In this study, both habitats—water and sediment—shared a core repertoire of DSR enzymes, including sulfate adenylyltransferase (*sat*, *met3*), adenylyl phosphosulfate reductase (*aprAB*), and dissimilatory (bi)sulfite reductase (*dsrAB*) (Figure 6a). These genes were primarily associated with taxa belonging to α-, γ, and *δ*-Proteobacteria, such as *Thiocapsa*, Sulfitobacter, *Halochromatium*, *Thioclava*, and *Pseudorhodobacter* in the water column, and Desulfobacteraceae_uc, *Desulfosarcina*, *Desulfococcus*, *Desulfotignum*, *Desulfobacter*, *Thiocapsa*, *Halochromatium*, *Sulfitobacter*, and *Pseudorhodobacter* in sediments, underscoring their universal role in DSR across these niches. Interestingly, *sat*, *met3* was significantly enriched in the water column compared to sediments, suggesting options for potential adaptations to dynamic environmental conditions where rapid sulfate activation might be advantageous. However, a notable finding was the absence of critical auxiliary components in the DSR pathway [47]. Key complexes such as *qmoAB(C)* and *dsrMK(JOP*), which facilitate electron transfer to *aprAB* and *dsrC* and are integral to the efficiency of sulfide production [4,12], were not detected in either habitat. Similarly, *dsrC*, which catalyzes the terminal reduction step releasing sulfide, was undetected, raising intriguing questions about how DSR is functionally sustained in these environments without these auxiliary components. This observation suggests the possibility of alternative electron transfer mechanisms or complementary metabolic pathways compensating for the absence of these complexes, warranting further investigation.

In summary, this study proposes a conceptual model that delineates the role of major microbial taxa in the mediation of S, C, and N biogeochemical cycles across the water column and sedimentary environments of the Isinuka sulfur pool (Figure 6b). In the water column, *Thiocapsa* functions as the primary phototrophic sulfur oxidizer (pathway 2, Figure 6b), facilitating the conversion of reduced sulfur compounds while concurrently linking sulfur and carbon cycling. Additionally, *Thiocapsa* spp. contributes to nitrate reduction, integrating nitrogen cycling processes and influencing nitrogen availability in aquatic systems [65]. This inference is based on genomic evidence from *T. bogorovii* BBS, which, despite its inability to grow with nitrate as a sole nitrogen source and its lack of key genes for assimilatory nitrate reduction (e.g., ferredoxin–nitrate reductase and nitrate reductase (NAD(P)H)), possesses genes encoding enzymes implicated in dissimilatory nitrate reduction and partial denitrification. Specifically, *T. bogorovii* BBS contains genes for nitrate reductase/nitrite oxidoreductase (alpha subunit) and nitrite reductase (cytochrome c-552), suggesting potential involvement in the conversion of nitrate to nitrite under anaerobic conditions [65]. Nevertheless, further experimental investigations are warranted to confirm the functional expression and ecological relevance of these pathways.

Within sediments, *Delftia* may play a pivotal role in multiple biogeochemical transformations. It actively participates in organic matter degradation, a key component of the carbon cycle, while also engaging in biological nitrogen fixation and nitrogen removal, thereby enhancing nitrogen assimilation in anoxic conditions [73,74]. Although *Delftia* is not a canonical sulfate reducer, its presence in sulfur-enriched environments and its genomic repertoire suggest the potential for partial sulfur cycling [75,76,77]. Spanning both the water column and sediments, *Lutimaribacter* serves as a functional bridge between sulfur and nitrogen cycling. It facilitates aerobic sulfur oxidation, contributing to sulfur turnover, while simultaneously engaging in denitrification, a critical pathway for nitrogen removal from aquatic systems. Collectively, these microbial interactions establish a tightly coupled network of sulfur, carbon, and nitrogen cycling, underpinning ecosystem stability and biogeochemical fluxes in sulfur-enriched aquatic environments.

Finally, the potential for human contamination in the Isinuka sulfur-rich therapeutic pool, used for balneotherapy and pelotherapy [7], warrants careful consideration from both a microbial ecology and public health perspective. While these pools are often rich in beneficial microbes, human activities, such as bathing, can introduce new microbial species or disrupt the existing microbial community. In this study, we identified an unusually high abundance of unclassified Coxiellaceae (8.028%) in the sediments of the sulfur-rich pool. Although Coxiellaceae are typically not abundant in sulfur-rich ecosystems, their presence may be influenced by human contamination, as bathers could introduce microorganisms via skin flora, sweat, or other metabolic byproducts. The specific environmental conditions within the sulfur pool, including high sulfur concentrations and low oxygen levels, may further provide a niche that favors the growth of Coxiellaceae, particularly those introduced by human activity. Moreover, the microbial community disturbance caused by human exposure could create a favorable environment for the proliferation of these taxa, thereby enhancing their relative abundance in the sediment. However, further investigations are needed to better understand the ecological role of Coxiellaceae in such environments and to assess the potential impacts of human activity on both the microbial community structure and function, as well as public health within therapeutic sulfur pools like Isinuka.

## 4. Conclusions

The Isinuka sulfur pool is a unique geochemical and microbial ecosystem characterized by distinct physicochemical parameters, microbial diversity, and metabolic processes that underpin sulfur, carbon, and nitrogen cycling. The findings reveal significant differences between the water column and sediment in terms of salinity, alkalinity, and nutrient concentrations, with sediments acting as reservoirs of solutes and supporting more diverse microbial communities. The microbial diversity, as indicated by 16S rRNA sequencing, was markedly higher in sediments, where redox gradients and resource heterogeneity foster complex microbial niches. Core microbiome analysis identified *Thiocapsa*, *Lutimaribacter*, and *Delftia* as functional keystones, integrating sulfur oxidation and nutrient recycling. Key microbial processes included sulfur oxidation in the water column dominated by *Thiocapsa* and other sulfur-oxidizing bacteria (SOB), while sulfate-reducing bacteria (SRB), such as *Desulfobacteraceae* and *Desulfococcus*, thrive in the sediment under anaerobic conditions. This stratification reflects niche-specific adaptations and functional roles in sulfur cycling. The interplay between SOB and SRB facilitates syntrophic interactions, ensuring efficient sulfur turnover and ecosystem stability. Functional predictions indicated that microbial communities in sediments excel in dissimilatory sulfate reduction, methane metabolism, and nitrogen fixation, while those in the water column focus on sulfur oxidation, cofactor biosynthesis, and denitrification. Biogeochemical processes are further enriched by carbon and nitrogen cycling. Carbon fixation, particularly through the rTCA cycle, and methane oxidation are vital in sediments, linking carbon to sulfur dynamics. Nitrogen cycling processes such as assimilatory nitrate reduction, sulfur autotrophic denitrification, and dissimilatory nitrate reduction to ammonia (DNRA) highlighted the interconnectedness of sulfur and nitrogen metabolism, contributing to nutrient balance and energy flow within this sulfur-rich environment.

This study’s limited sample size and single temporal sampling constrain its capacity to capture the dynamic and multifaceted nature of microbial community structure and function within the Isinuka sulfur pool. The reliance on PICRUSt-based functional predictions introduces inherent uncertainties, as such in silico approaches may not accurately represent in situ metabolic activities. Furthermore, the absence of direct functional data, such as metagenomic or metatranscriptomic analyses, impedes the ability to draw definitive conclusions regarding the ecological mechanisms driving sulfur cycling in this system. Despite these constraints, this study offers significant contributions to understanding sulfur metabolism in atypical, sulfur-enriched environments. The findings underscore the Isinuka sulfur pool’s utility as a model system for investigating microbial diversity and sulfur cycling under unique geochemical conditions. Future research incorporating longitudinal sampling, high-resolution multi-omics techniques, and direct functional profiling will address these limitations, providing a more robust characterization of the system’s ecological and biogeochemical dynamics with implications for biotechnological applications, including bioremediation and sulfur-based biofertilizer development. Additionally, future studies incorporating isotopically labeled substrates (e.g., ^13^C-labeled organic compounds, ^15^N-labeled nitrate, and ^34^S-labeled sulfate) are needed to track microbial metabolic fluxes and quantify the contributions of each microbial taxon to sulfur oxidation, nitrogen transformation, and organic matter turnover.

## Figures and Tables

**Figure 1 biology-14-00503-f001:**
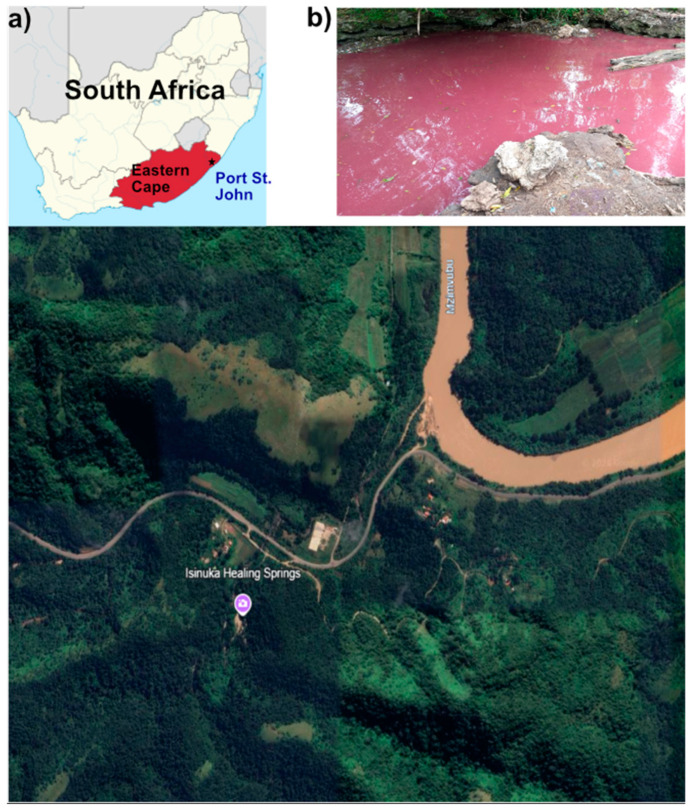
Location of the study site at Isinuka Healing Springs. (**a**) Map of South Africa showing the location of Port St. Johns (*) in the Eastern Cape Province, the nearest town to Isinuka Healing Springs. Lower image shows an aerial view of the spring and surrounding landscape (image source: Google Earth, Maxar Technologies). (**b**) Photograph of the bathing pond at Isinuka Healing Springs, where environmental sampling of water and sediment was conducted.

**Figure 2 biology-14-00503-f002:**
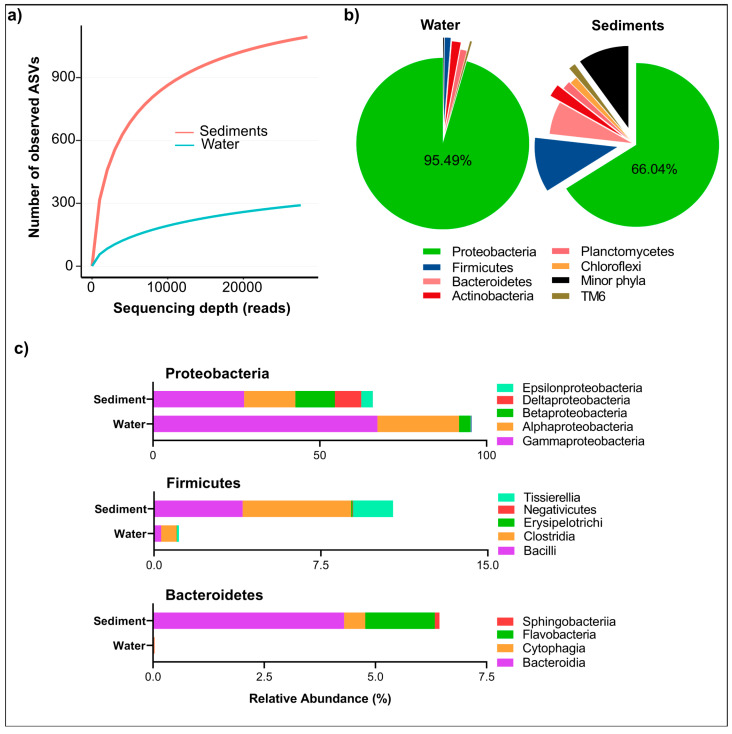
Microbial diversity and alpha rarefaction of bacterial community in water and sediments from the Isinuka sulfur pool. (**a**) Rarefaction curves constructed based on ASVs at a 3% dissimilarity level, illustrating the diversity of bacterial communities in water and sediment samples. (**b**) Pie chart depicting the distribution of microbial communities at the phylum level for both water and sediment samples. (**c**) Bar plots representing the major classes within the top three bacterial phyla identified in water and sediment samples.

**Figure 3 biology-14-00503-f003:**
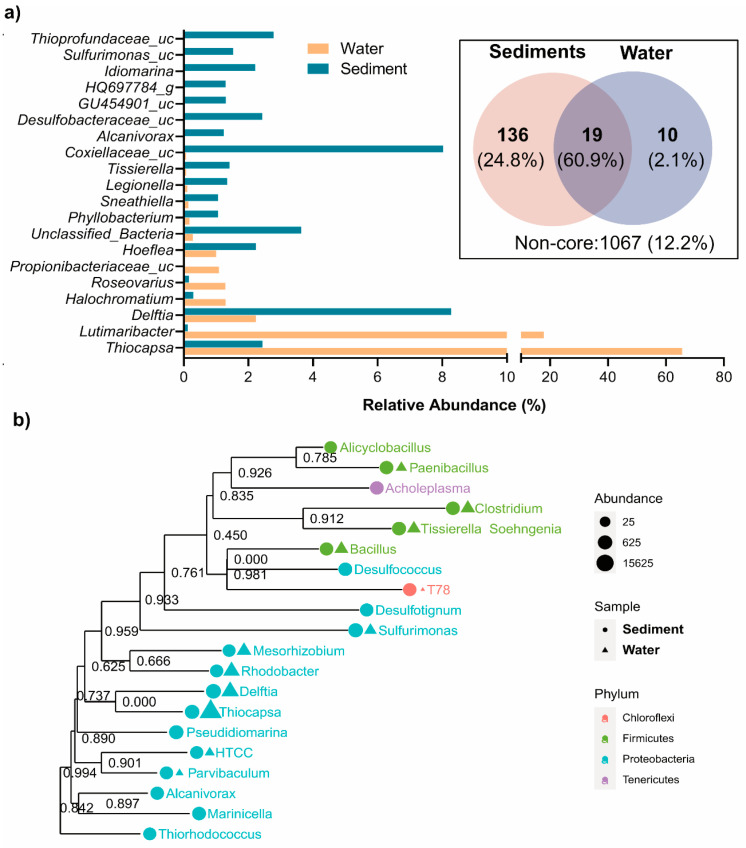
Diversity of bacterial communities in water and sediment samples of the Isinuka sulfur pool. (**a**) Top taxa at genus level identified. The inset depicts a Venn diagram illustrating the shared and unique core microbiomes at the genus level, defined as ASVs present in at least 50% of the samples in each group with a minimum relative abundance of 0.5%. (**b**) Maximum likelihood phylogenetic tree of the key genera, with their abundances in water and sediment samples. Bootstrap values of the branches are provided in the phylogenetic tree.

**Figure 4 biology-14-00503-f004:**
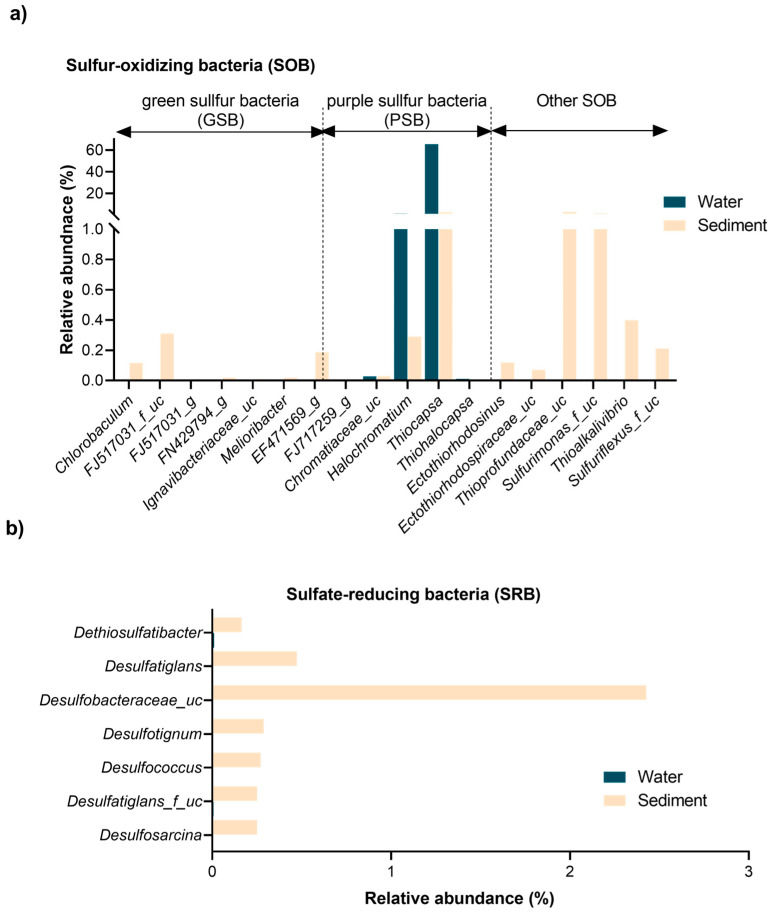
Distribution and diversity of major sulfur-metabolizing bacterial taxa in the Isinuka sulfur pool. (**a**) Sulfur-oxidizing bacteria (SOB), highlighting the abundance of both anoxygenic photosynthetic purple sulfur bacteria (PSB) and green sulfur bacteria (GSB). (**b**) Dominant sulfate-reducing bacteria (SRB), detected exclusively in sediment samples.

**Figure 5 biology-14-00503-f005:**
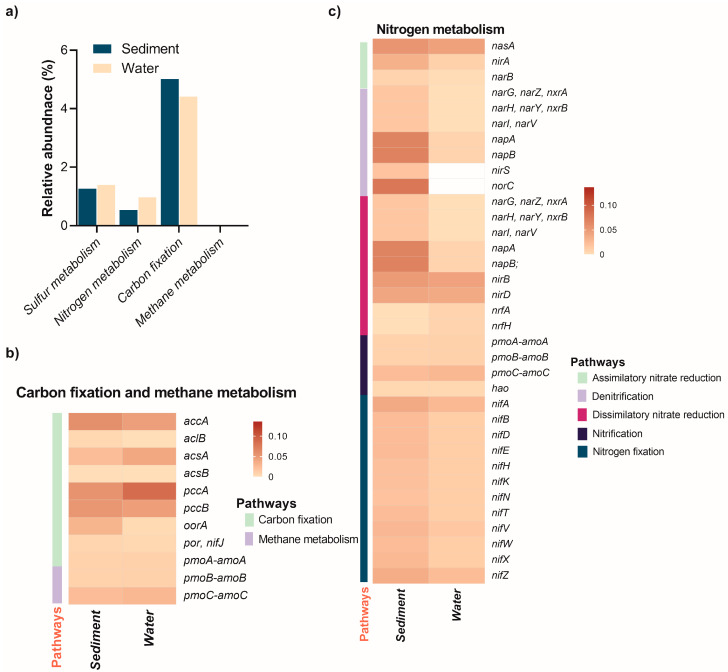
Relative abundance of elemental metabolism pathways and associated energy metabolism genes in the sediment and water of the Isinuka sulfur pool, as identified through KEGG orthologs using PICRUSt2 analysis. (**a**) Comparative analysis of C, N, and S metabolism pathways across the two niches. (**b**,**c**) Heatmap illustrating the distribution of key genes involved in energy metabolism (C and N, and S) in both habitats, with the color bar representing the relative abundance of genes. Detailed information on the abundance of C and N metabolism genes is provided in Appendix A, respectively.

**Figure 6 biology-14-00503-f006:**
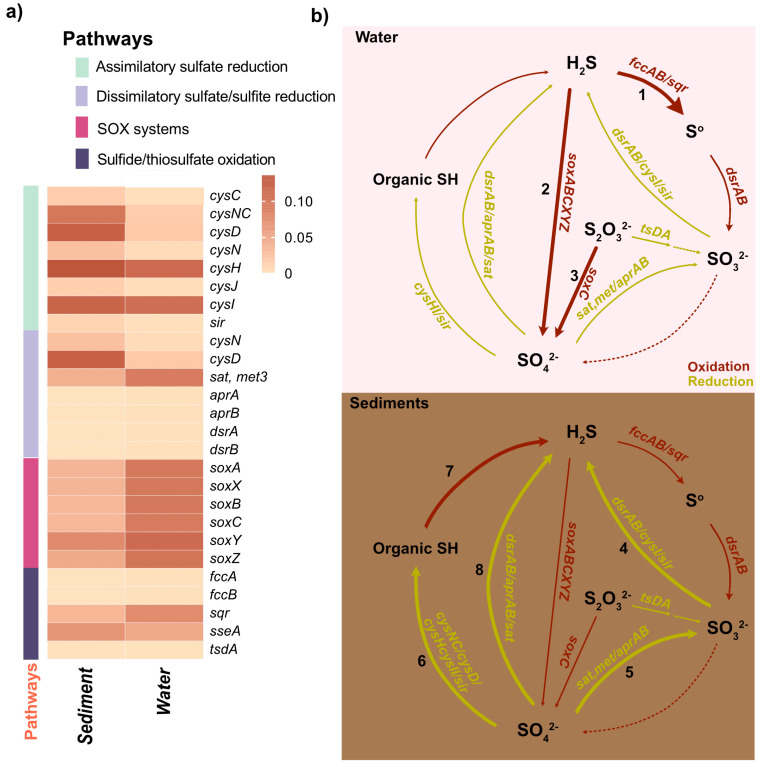
Partitioning of sulfur metabolism in the water column and sediments of the Isinuka sulfur pool. (**a**) Heatmap displaying key genes involved in sulfur metabolism pathways predicted in water and sediment samples. The color bar represents the relative abundance of these genes. Detailed information on the abundance of sulfur metabolism genes is provided in Appendix A. (**b**) Diagram illustrating sulfur metabolic pathways and associated genes enriched in the water column and sediment compartments. Dominant pathways in the water column include: (1) sulfide oxidation; (2) anoxygenic photosynthesis; and (3) thiosulfate oxidation. In sediments, predominant pathways include: (4) dissimilatory sulfite reduction (DSR); (5) dissimilatory sulfate reduction to sulfite; (6) assimilatory sulfate reduction (ASR); (7) desulfurylation/mineralization; and (8) dissimilatory sulfate reduction (DSR). Associated microbial taxa for each pathway are discussed in the main text. The thickness of each line reflects the enrichment of pathway genes, while dashed lines indicate pathways that were not predicted.

**Table 1 biology-14-00503-t001:** The physicochemical parameters of sediment and water samples from the Isinuka sulfur pool ^†^.

	Sediment (mg/kg)	Water (mg/L)	*p*-Value
**Physicochemical parameters**			
pH	8.6 ± 0.69	8.1 ± 0.52	0.3363
Temperature (^o^C)	16.6 ± 0.82	17.3 ± 1.04	0.0985
Salinity	5.0 ± 0.32	3.0 ± 0.08	<0.001 ***
Alkalinity	518 ± 36	384 ± 21	0.0052 **
Total solids (TS)	3470 ± 68	1958 ± 134	<0.001 ***
Soluble solids (SS)	44 ± 3.29	88 ± 4.56	0.0002 ***
Total dissolved solids (TDS)	3424 ± 57	1879 ± 125	<0.0001 ***
Dissolved oxygen (DO)	0.0	0.27 ± 0.107	0.0025 **
NH_4_^+^	8.8 ± 1.77	1.56 ± 0.94	0.0034 **
NO_3_^−^	3.3 ± 0.23	0.71 ± 0.26	0.0002 ***
NO_2_^−^	0.63 ± 0.06	0.46 ± 0.13	0.1221
Cl^−^	1192 ± 39	686.36 ± 59	0.0002 ***
SO_4_^2−^	139 ± 11	77 ± 6.71	0.0013 **
SO_3_^2−^	1.72 ± 0.33	0.34 ± 0.16	0.0031 **
H_2_S	4.98 ± 2.05	15.3 ± 3.16	0.0154 *
**Trace metals and metalloids**			
As	7.52 ± 0.49	25 ± 3.71	0.0012 **
Ca	11,196 ± 38	11,478 ± 94	0.0086
Cd	49 ± 2.69	33 ± 5.43	0.0096 **
Co	46 ± 8.97	26 ± 4.50	0.0258*
Cr	162 ± 123	49 ± 1.39	0.1878
Cu	53 ± 7.72	25 ± 3.02	0.004 **
Fe	24,093 ± 1312	190 ± 16	0 ***
Mg	8775 ± 476	14,767 ± 579	0.0002 ***
Mn	7109 ± 497	50 ± 4.54	0 ***
Mo	38 ± 3.19	68 ± 82.06	0.5644
Ni	145 ± 16.85	33 ± 5.39	0.0004 ***
Pb	20 ± 12.60	15 ± 3.21	0.5833
Se	5.9 ± 3.21	51 ± 7.77	0.0007 ***
Zn	184 ± 39	17 ± 5.40	0.0018 **

^†^ Values represent mean and standard deviation of 5 samples for both sediment and water. Superscripts beside *p*-values are significantly different measures (*p* ≤ 0.05) based on Fisher’s least significant difference. (Significance codes: <0.001 ‘***’ 0.001 ‘**’ 0.01 ‘*’ 0.05.)

**Table 2 biology-14-00503-t002:** Summary of sequencing data and alpha diversity indices of the bacterial microbiome of water and sediment samples from Isinuka sulfur pool.

Parameter	Water	Sediment
Quality filtered reads classified	27,565	28,451
Unique ASVs	291	1094
Shannon	1.86	5.25
Simpson	0.73	0.98
invSimpson	3.66	41.8
ACE	396.2	1204.2
Chao1	403.0	1199.0
Good’s coverage (%)	99.34	99.90

## Data Availability

The original contributions presented in this study are included in the article/Appendix A.

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
