# Peer review of "Thiocapsa*, *Lutimaribacter*, and *Delftia* Are Major Bacterial Taxa Facilitating the Coupling of Sulfur Oxidation and Nutrient Recycling in the Sulfide-Rich Isinuka Spring in South Africa"

_biology, 2025, doi:10.3390/biology14050503_

Round 1

Reviewer 1 Report

Comments and Suggestions for Authors

Dear Authors,

The content of the manscipt looks quite fine and reasonable.

I have mostly comments on some details of the manuscript:

30L - I do not agree to write "sulfur-reducing microbes such as Desulfobacteraceae"

50L - Desulfobacteraceae - it is worth to mention that it was "unclassified" type.

65L - with oxidized forms like - perhaps, with oxidized form like ...

84 - "dark grey water and the striking pink hue of the sulfur pool" - pink color could be due to the dominance of purple anoxygenic phototrophs - what is in your case? COuld it be some pink precipitates?

98L - "Central to this cycle are key microbial processes, including sulfide oxidation (SOX), sulfate reduction, and elemental sulfur transformations [1,4,8]. Sulfur-oxidizing bacteria (SOB), such as Thiobacillus, Beggiatoa, and Acidithiobacillus, oxidize hydrogen sulfide (H₂S) to sulfate (SO₄²⁻) or elemental sulfur (S⁰), contributing to the system's sulfur chemistry [2,9]." - could it be better to modify (or not?) - ... including phototrophic and/or chemotrophic sulfide oxidation ...  and add some phototrophs

Table 1. Please, provide necessary charges to all compounds that have charge! Metal ions do have to have also! Especially that is important for ions with varying charges (Fe & Mn). COncerning Fe it has to be described and explained what kind of Fe was measured! What is SS in the Table?

256 - DO level (0.27 ± 1.07 mg/L). Please, check calculation - it is not possible to measure negative concentration of DO!

263 - This pattern suggests active sulfate reduction in the sediments - not exactly: This pattern suggests for the possibility !!! for active sulfate reduction in the sediments

330 - Composition of bacterial composition - what is that? please, correct!

 449, 472 - SRB - that is ofter associated with sulfate reducers also within this manucript 323, 765 ... - "These sulfate-reducing bacteria (SRB) have ... = please, clear modify to distinguish S-reducing and sulfate-redicung!

FIg 4B - Please, provide a proof (reference) that Magnetovibrio is S-reducer. Why all sulfate-reducers are shown here as S-resucer in sediment? If sulfate was present - they would function as sulfate-reducers.

590 - These genes are likely contributed by methane-oxidizing bacteria belonging to α-proteobacteria, and γ-proteobacteria in anaerobic, sulfur-rich habitats [59]. - Please, discuss what aerobic bacteria could do in anaerobic S-rich environment.

620 - Please, prove that Nitrosomonas and Azoarcus are autotrophs that could denitrify with S! Desulftobacter - ?

657 - unclassified Desulfobacteraceae ... well-documented for sulfur oxidation and nitrate reduction - Please, check this point and provide reference.

Fig 6

a. DSR belongs to SOX? 6b - probably it is more correct: DSR - sulfite to sulfide, rDSR - sulfide to sulfite? Please, check colors in the 6b - seems that they do not correspond.

Seems that many mistakes in pathways description: sulfate oxidation ... , assimilatory/dissimilatoty sulfur reduction using APS ...?

Probable you have to indicate where you meant sulfur as a general name fo inorganic compounds containing atom of S, and where you meant that sulfur is So (zero charge - elemental sulfur).

756 - assimilatory sulfate reductases - what kind of genes that were?

774 powered by sulfite/sulfate reductases (Dsr) - Dsr are responcible for sulfite but not sulfate reduction.

810 Furthermore, Delftia is involved in sulfate reduction, leading to the production of hydrogen sulfide (H₂S), which modulates sulfur cycling and sediment redox dynamics. - Please, prove that Deslftia spp that are described and known as aerobic bacteria are sulfate reducers under anoxic conditions!

905 - not correct name of the reference

References. There were some incorrect and not precise citations of environmental papers (Delftia for ex.) Also when you stated that some representative of SRB are active sulfur reducers - please prove that  with exact references.

Author Response

Comment 1. 30L - I do not agree to write "sulfur-reducing microbes such as Desulfobacteraceae"

Response 1. In this study, we identified unclassified Desulfobacteraceae as one of sulfate reducing bacteria. We have thus corrected the sentence in line 30 to correctly reflect this “…………..while sediments hosted sulfate-reducing microbes in the family Desulfobacteraceae, adapted to low-oxygen environments

Comment 2. 50L - Desulfobacteraceae - it is worth to mention that it was "unclassified" type.

Response 2. We have correctly rectified this as suggested by reviewer in line 49

Comment 3. 65L - with oxidized forms like - perhaps, with oxidized form like ...

Response 3. We have revised accordingly in line 65 as suggested by reviewer to read “……..₃²⁻), along with oxidized form like sulfate (SO₄²⁻), which

Comment 4. 84 - "dark grey water and the striking pink hue of the sulfur pool" - pink color could be due to the dominance of purple anoxygenic phototrophs - what is in your case? COuld it be some pink precipitates?

Response 4. The sentence has been restructure to bring clarity in line 83-84 to read “………, dark grey sediments (mud), the striking pink hue of the sulfur pool linked to enriched phototrophic purple sulfur bacteria (PSB) and the presence of bubbling H₂S,…………”

Comment 5. 98L - "Central to this cycle are key microbial processes, including sulfide oxidation (SOX), sulfate reduction, and elemental sulfur transformations [1,4,8]. Sulfur-oxidizing bacteria (SOB), such as Thiobacillus, Beggiatoa, and Acidithiobacillus, oxidize hydrogen sulfide (H₂S) to sulfate (SO₄²⁻) or elemental sulfur (S⁰), contributing to the system's sulfur chemistry [2,9]." - could it be better to modify (or not?) - ... including phototrophic and/or chemotrophic sulfide oxidation ...  and add some phototrophs

Response 5. We appreciate the reviewer’s suggestion regarding the inclusion of phototrophic sulfide oxidation to provide a more comprehensive representation of sulfur cycling in the Isinuka sulfur pool. In response, we have revised the section to explicitly distinguish between phototrophic and chemotrophic sulfide oxidation, highlighting their respective microbial taxa (line 98-102). This modification enhances clarity and ensures that both metabolic pathways contributing to sulfur transformations are adequately represented. The revised text now states: "Central to this cycle are key microbial processes, including phototrophic and chemo-trophic sulfide oxidation, sulfate reduction, and elemental sulfur transformations [1,4,8]. Phototrophic sulfur-oxidizing bacteria (e.g., Thiocapsa, Chromatium, Chlorobium) and chemotrophic sulfur oxidizers (e.g., Thiobacillus, Beggiatoa, Acidithiobacillus) drive the oxidation of H₂S to SO₄²⁻ or S⁰, contributing to the system’s sulfur chemistry [2,9]."

Comment 6: Table 1. Please, provide necessary charges to all compounds that have charge! Metal ions do have to have also! Especially that is important for ions with varying charges (Fe & Mn). COncerning Fe it has to be described and explained what kind of Fe was measured! What is SS in the Table?

Response 6: Thank you for your insightful comment. We acknowledge the importance of oxidation states in understanding the geochemical behavior of Fe and Mn. However, as stated in our methodology, we utilized Inductively Coupled Plasma Optical Emission Spectroscopy (ICP-OES) and Inductively Coupled Plasma Mass Spectrometry (ICP-MS), which do not directly measure oxidation states. These techniques determine the total elemental concentration of Fe and Mn in the sample, irrespective of their oxidation states.

Comment 7. 256 - DO level (0.27 ± 1.07 mg/L). Please, check calculation - it is not possible to measure negative concentration of DO!

Response 7. Thank you for your observation. We acknowledge the typographical error in the reported dissolved oxygen (DO) values. The correct values are 0.27 ± 0.107 mg/L, not 0.27 ± 1.07 mg/L. This correction has been made accordingly in Table 1 and the main text (line 288) to ensure accuracy and consistency. We appreciate the reviewer’s attention to detail.

Comment 8. 263 - This pattern suggests active sulfate reduction in the sediments - not exactly: This pattern suggests for the possibility !!! for active sulfate reduction in the sediments

Response 8. This has been corrected accordingly in line 301 to read “This pattern suggests possibility of active SO4²⁻ reduction in the sediments,……………..”

Comment 9. 330 - Composition of bacterial composition - what is that? please, correct!

Response 9. This subsection title has been corrected to read has been corrected accordingly in line 301 to read “3.3 Taxonomic composition of bacterial community

 Comment 10. 449, 472 - SRB - that is ofter associated with sulfate reducers also within this manucript 323, 765 ... - "These sulfate-reducing bacteria (SRB) have ... = please, clear modify to distinguish S-reducing and sulfate-redicung!

Response 10. We appreciate the reviewer’s attention to the important distinction between sulfur-reducing and sulfate-reducing bacteria. In the revised manuscript, we have carefully reviewed and modified the relevant sections (lines 323, 449, 472, 765) to ensure accurate terminology. Specifically, we now distinguish between:

  1. Sulfate-reducing bacteria (SRB): microbes that reduce sulfate (SO₄²⁻) to sulfide (H₂S), typically under anaerobic conditions (e.g., Desulfobacteraceae, Desulfosarcina).
  2. Sulfur-reducing bacteria: microbes that reduce elemental sulfur (S⁰) or polysulfides to sulfide, independent of sulfate as an electron acceptor (e.g., Dethiosulfatibacter and other taxa in sulfur-enriched environments).

Where applicable, references to "SRB" have been explicitly linked to sulfate reduction, and organisms involved in elemental sulfur reduction are now clearly described as sulfur-reducing bacteria. This distinction has also been reflected in the discussion of microbial functional roles and ecological implications of sulfur transformations

Comment 11. FIg 4B - Please, provide a proof (reference) that Magnetovibrio is S-reducer. Why all sulfate-reducers are shown here as S-resucer in sediment? If sulfate was present - they would function as sulfate-reducers.

Response 11. We acknowledge the erroneous classification of Magnetovibrio as a sulfur-reducing bacterium in Figure 4B. Upon re-evaluation, we found insufficient evidence in the current literature to support its functional role in sulfur or sulfate reduction. Accordingly, Magnetovibrio has been removed from the figure to ensure taxonomic and functional accuracy. Additionally, we recognize the need for greater clarity in the categorization of sulfate-reducing versus sulfur-reducing bacteria in the figure. The original presentation may have conflated these groups. In response, we have revised the figure legend and relevant text to explicitly denote that the taxa depicted under “sulfur reduction” primarily represent sulfate-reducing bacteria (SRB), as sulfate was the major sulfur species present in the sediment samples (as confirmed by geochemical analyses). This correction aligns the functional annotation with the environmental context and observed geochemical conditions.

Comment 12. 590 - These genes are likely contributed by methane-oxidizing bacteria belonging to α-proteobacteria, and γ-proteobacteria in anaerobic, sulfur-rich habitats [59]. - Please, discuss what aerobic bacteria could do in anaerobic S-rich environment.

Response 12. We thank the reviewer for highlighting the need to clarify the apparent contradiction between aerobic methanotrophs and their occurrence in anaerobic, sulfur-rich environments. To address this, we have modified the sentence in line 637-639 to read:

“These genes are likely contributed by methane-oxidizing bacteria belonging to α- and γ-proteobacteria which—though classically aerobic—are known to persist in microaerophilic or anoxic environments by employing fermentation-based methanotrophy or facultative anaerobic pathways such as denitrification [62,63].”

To further substantiate this clarification, we have included an additional reference (Schorn et al., 2024), which provides recent evidence of aerobic methanotrophs surviving under anoxic lacustrine conditions by utilizing alternative metabolic strategies such as fermentation-based methanotrophy and denitrification. This supports the ecological plausibility of their presence and activity in the anaerobic, sulfur-rich habitats described in our study.

Comment 13. 620 - Please, prove that Nitrosomonas and Azoarcus are autotrophs that could denitrify with S! Desulftobacter - ?

Response 13. The sentence has been revised for clarity and accuracy in line 667-672. It now reads:

The key taxa enriched in the sediment samples that have been previously reported to undertake sulfur-driven autotrophic denitrification (SADN) under diverse environments include Paracoccus, Sulfurimonas, Pseudorhodobacter, Thioalkalivibrio, Rhodobacter, and Halomonas [11,12,66–68]. While Nitrosomonas and Azoarcus are not classical SADN organisms, emerging evidence suggests their potential or indirect involvement under specific environmental conditions—such as denitrification-linked sulfur oxidation or facultative interactions with sulfur compounds [67,68].”

This revision acknowledges the reviewer's concern regarding the roles of Nitrosomonas, Azoarcus, and Desulfobacter, and limits the list of classical SADN taxa to those with well-established functions. Supporting references (Lee et al., 2014; Jiang et al., 2025) have been included to substantiate the emerging roles of Nitrosomonas and Azoarcus in sulfur-associated nitrogen cycling.

Comment 14. 657 - unclassified Desulfobacteraceae ... well-documented for sulfur oxidation and nitrate reduction - Please, check this point and provide reference.

Response 14. Thank you for pointing this out. Upon careful review, we acknowledge that Desulfobacteraceae are traditionally recognized as sulfate-reducing bacteria (SRB), primarily involved in the reduction of sulfate to sulfide under anoxic conditions. As such, they are not typically associated with sulfur oxidation. In light of this, and to avoid redundancy with the information presented in lines 667–672, we have revised the manuscript accordingly by removing the statement regarding their involvement in sulfur oxidation and nitrate reduction.

Comment 15. Fig 6. DSR belongs to SOX? 6b - probably it is more correct: DSR - sulfite to sulfide, rDSR - sulfide to sulfite? Please, check colors in the 6b - seems that they do not correspond.

Response 15. Thank you for your insightful observation. We agree that the representation of dissimilatory sulfur metabolism required clarification. In response, we have revised and simplified the sulfur cycle diagram in Figure 6b to more accurately reflect the directionality of key sulfur transformations. Specifically, we now distinguish between dissimilatory sulfite reduction (DSR)—i.e., the reduction of sulfite to sulfide—and reverse DSR (rDSR), which mediates sulfide oxidation to sulfite, depending on environmental conditions and microbial context. Additionally, the color coding in the revised figure has been adjusted to ensure accurate correspondence with the legend and improve interpretability. The updated schematic now emphasizes the predominant pathways enriched in the water column and sediments, with consistent visual alignment between pathway lines, genes, and associated compartments.

Comment 16. Seems that many mistakes in pathways description: sulfate oxidation ... , assimilatory/dissimilatoty sulfur reduction using APS ...?

Response 16. We have revised and simplified the sulfur cycle diagram in Figure 6b to more accurately reflect the directionality of key sulfur transformations.

Comment 17. Probable you have to indicate where you meant sulfur as a general name fo inorganic compounds containing atom of S, and where you meant that sulfur is So (zero charge - elemental sulfur).

Response 17. We acknowledge the potential ambiguity in the manuscript regarding the use of the term sulfur. We have now carefully reviewed the text and clarified where sulfur refers to elemental sulfur (S⁰) versus where it is used more broadly to represent inorganic sulfur species (e.g., sulfide, sulfite, sulfate). These distinctions have been explicitly indicated in relevant sections of the text to improve clarity and scientific accuracy.

Comment 18. 756 - assimilatory sulfate reductases - what kind of genes that were?

Response 18. We have modified the sentence to provide more clarity and explicitly identify the specific genes involved. In line 801-804, the sentence has been revised to:

"In this study, sediments exhibited higher levels of genes such as sulfate adenylyltransferase and adenylylsulfate kinase (cysNC), sulfate adenylyltransferase subunit 2 (cysD), PAPS reductase (cysH), and sulfite reductase (cysI) (Figure 5d), suggesting a diverse and robust ASR system.",

Comment 19. 774 powered by sulfite/sulfate reductases (Dsr) - Dsr are responcible for sulfite but not sulfate reduction.

Response 19. We acknowledge that Dsr enzymes are indeed responsible for sulfite reduction rather than sulfate reduction. To correct this, we have revised the sentence to reflect this distinction more accurately. The sentence has been modified in line 817-823 and throughout the manuscript where applicable.

Comment 20. 810 Furthermore, Delftia is involved in sulfate reduction, leading to the production of hydrogen sulfide (H₂S), which modulates sulfur cycling and sediment redox dynamics. - Please, prove that Deslftia spp that are described and known as aerobic bacteria are sulfate reducers under anoxic conditions!

Response 20. We appreciate the reviewer's concern regarding the statement about Delftia and sulfate reduction. We acknowledge that Delftia species are primarily known as aerobic bacteria. The statement was not intended to imply that Delftia are obligate or primary sulfate-reducing bacteria in the same way as dedicated anaerobic sulfate-reducing bacteria (SRB). Our intent was to highlight the potential role of Delftia within the broader context of sulfur cycling, particularly in environments where sulfate reduction is significant. We have revised the sentence to a more nuanced description of its potential role in sulfur cycling within these complex environments, and include the genomic information found. We have also added citations to support the genomic information, and the complex interactions of microbes within sulfur cycling environments.

Comment 21. 905 - not correct name of the reference

Response 21. The correct full title of the cited paper has been provided

Comment 22. References. There were some incorrect and not precise citations of environmental papers (Delftia for ex.) Also when you stated that some representative of SRB are active sulfur reducers - please prove that with exact references.

Response 22. Thank you for your valuable feedback. We have carefully reviewed the references and made the necessary corrections. In particular, we acknowledge that the citation of Delftia was not as precise as required.

Reviewer 2 Report

Comments and Suggestions for Authors

Review comments

This manuscript analyzed bacterial diversity and metabolic potential in sediment and water samples. It has significant theoretical research significance. However, the experimental design is simple, and it still needs further revision and improvement. The specific comments are as follows.

  1. When abbreviations first appear, please use the full name, and use abbreviations afterwards. Please check and modify duplicate abbreviations. Like 101, oxidize hydrogen sulfide (H₂S) to sulfate (SO₄²⁻) or elemental sulfur (S⁰), line 154, and so on.
  2. Line 132, abbreviations that first appear require the full name.
  3. There are many writing errors in the manuscript, please carefully check. Like 158, H₂S; line 229, and so on.
  4. Table 1, in Physicochemical parameters, NH4, NO3, NO2, whether they are in ionic form needs to be clearly stated.
  5. Heavy metals, As and Mg do not belong to heavy metals.
  6. Lines 144 and 146, “Fig. ”, and the others are “Figure”. Please unify.
  7. The bacterial name in the picture needs to be italicized.
  8. Line 412, “(23) and (24)”, are they literatures? Please check the format.
  9. The discussion is not thorough enough, and it is suggested to explain the potential significance.
Comments on the Quality of English Language

It can be understood, but the language expression is not refined enough.

Author Response

Reviewer 1

Comment 1: This manuscript analyzed bacterial diversity and metabolic potential in sediment and water samples. It has significant theoretical research significance. However, the experimental design is simple, and it still needs further revision and improvement. The specific comments are as follows.

Response 1. We sincerely appreciate the reviewer's valuable feedback regarding methodological clarity. We have comprehensively revised the Materials and Methods section to include full experimental details, addressing both sampling protocols and analytical procedures such as enhanced enhanced sampling protocol (Section 2.1 line 172-186) and detailed physicochemical analyses (Section 2.2 line 188-211).

Comment 2: When abbreviations first appear, please use the full name, and use abbreviations afterwards. Please check and modify duplicate abbreviations. Like 101, oxidize hydrogen sulfide (H₂S) to sulfate (SO₄²⁻) or elemental sulfur (S⁰), line 154, and so on.

Response 2. We sincerely appreciate the reviewer's careful attention to abbreviation consistency. We have systematically revised the manuscript to ensure all abbreviations follow the prescribed format: full term at first mention followed by the abbreviation in parentheses, with subsequent uses of the abbreviation only (Line 100, 102). All duplicate abbreviations have been corrected.

Comment 3: Line 132, abbreviations that first appear require the full name.

Response 3. The full name for PICRUSt (Phylogenetic Investigation of Communities by Reconstruction of Unobserved States) has been provided in line 131-132

Comment 4: There are many writing errors in the manuscript, please carefully check. Like 158, H₂S; line 229, and so on.

Response 4. This has been corrected accordingly. Line 157-159 now reads “The continuous release of H₂S is evident through bubbling activity within the pond, and additional gas fissures emitting H₂S are present approximately 20 m from the pool, contributing to the site's characteristic sulfurous odor [6,7]

Line 229 now 265 reads “PICRUSt2 generates a predictive metagenome by integrating ASV taxonomic classifications with reference genome data, enabling functional inferences about the microbial community [25].”

Comment 5: Table 1, in Physicochemical parameters, NH4, NO3, NO2, whether they are in ionic form needs to be clearly stated.

Response 5. We have revised Table 1 to explicitly indicate the ionic forms (NH₄⁺, NO₃⁻, NO₂⁻)  of nitrogen compounds

Comment 6: Heavy metals, As and Mg do not belong to heavy metals.

Response 6. We sincerely thank the reviewer for this important clarification regarding elemental classification. We have implemented the following revisions to ensure precise terminology. The header update in Table has been changed from "Heavy Metals" to "Trace Elements and Metalloids". Methods Section Refinement (Lines 204-205) revised to read " Trace elements (including transition/heavy metals), metalloids (As), and light met-als (Al, Mg) in water and sediment samples ..............."

Comment 7: Lines 144 and 146, “Fig. ”, and the others are “Figure”. Please unify.

Response 7. This has been corrected accordingly in line 144 and 146

Comment 9: Line 412, “(23) and (24)”, are they literatures? Please check the format.

Response 9. This has been corrected in line 437440 to right citation to read “In addition, Lutimaribacter species, known for their capability to degrade cyclohexylacetate, have previously been isolated from seawater [43]. Recently, Somee et al [44] reported the enrichment of Lutimaribacter in sulfur-rich oil spills in the marine ecosystem of the Persian Gulf.”

Comment 10: The discussion is not thorough enough, and it is suggested to explain the potential significance.

Response 10. We appreciate thorough evaluation of our manuscript by the reviewer and the opportunity to clarify the sufficiency of our discussion regarding the ecological significance of the Isinuka sulfur pool microbiome. We respectfully assert that our current discussion comprehensively addresses microbial diversity patterns, environmental gradients, biogeochemical processes, and potential applications in ecosystem management. Specifically, our discussion effectively links microbial community composition to environmental heterogeneity, biogeochemical cycling, and niche differentiation.

  1. Our analysis captures the extensive microbial diversity spanning 54 phyla, 114 classes, 226 orders, and 920 genera, illustrating the structural complexity of the Isinuka sulfur pool microbiome. The differentiation between water and sediment communities underscores how environmental factors such as oxygen availability, resource richness, and habitat stability shape bacterial assemblages. This stratification is well contextualized within the broader framework of microbial ecology in extreme environments.

  2. We have integrated a functional perspective into the taxonomic analysis by discussing the ecological roles of key microbial groups. For instance, Thiocapsa, the predominant sulfur oxidizer in the water column, is highlighted for its role in anoxygenic photosynthesis and sulfur oxidation, linking primary production to sulfur transformations. Similarly, Delftia’s contribution to organic matter degradation and the role of sulfate-reducing bacteria (SRB) in anaerobic sulfur cycling are extensively discussed, ensuring that our study is not merely descriptive but functionally informative.

  3. The core microbiome analysis strengthens our ecological interpretation by identifying shared and niche-specific taxa that drive key biogeochemical processes. We explicitly discuss the interplay of sulfur, nitrogen, iron, and carbon cycling facilitated by these microorganisms, emphasizing how microbial interactions contribute to the dynamic equilibrium of the sulfur pool.

  4. The clear delineation of unique microbial taxa in sediment versus water highlights how environmental pressures drive niche differentiation. We provide evidence that sediment-associated microbiomes are enriched in anaerobic sulfur-reducing bacteria and organic matter degraders, which thrive in stable, resource-rich, anoxic conditions. In contrast, water-column microbiomes are dominated by sulfur-oxidizing bacteria adapted to fluctuating oxygen levels and dynamic nutrient availability.

  5. The Isinuka sulfur pool represents a unique geochemical and microbial ecosystem where microbial-driven transformations of carbon, nitrogen, and sulfur are tightly coupled. Our study explicitly demonstrates how these interconnected cycles regulate elemental fluxes, emphasizing the ecological and geochemical significance of this habitat. By linking microbial functional groups to biogeochemical processes, we provide a holistic understanding of ecosystem stability and function. Taken together, our discussion presents a holistic interpretation of the Isinuka sulfur pool microbiome, effectively bridging taxonomic, functional, and environmental perspectives. We believe that these insights sufficiently address ecological significance, microbial-environment interactions, and potential ecosystem applications.

Reviewer 3 Report

Comments and Suggestions for Authors

This manuscript investigated the microbial communities in the water and sediments of this sulfur-rich pool, focusing on their diversity, roles, and impact on sulfur and nutrient cycling by using advanced DNA sequencing technologies, and found that Thiocapsa, Lutimaribacter and Delftia are major bacterial taxa  facilitating coupling of sulfur oxidation and nutrient recycling in sulfide-rich Isinuka spring in South Africa. It is provides key theoretical insights into microbial dynamics in sulfur-rich environments, highlighting their roles in biogeochemical cycling and potential applications in environmental management. However, prior to being accepted, there are some issues that should be addressed.

General comments

  1. 1. The Material and Method Partneeds to be improved. There are many key methods that have not been mentioned.For examples, (1)How are sediment and water samples collected? What is their quantity? How many replicates were collected for each sample? (2)Many Physicochemical Parameters, such as NH4, and heavy metals, were mentioned in the results section (Table 1), but the author did not describe them in the methodology.
  2. 2. Lines 174-180, These sentences can be moved to the previous section (2.1).
  3. Line 197-198, Amplicon sequence variant (ASV) was used in the method, but operational taxonomic units (OTUs)(Line 292)was used in the results and discussion. Please provide a unified description.
  4. Table 1, The unit (mg/L or mg/Kg) indication in the table is unclear. “Mg/Kg”can be placed under sediment, and “mg/L”can be placed under water.
  5. Lines 292, 293, 351, 390, 396, 397, ASV should be uniformly used here instead of OTU.
  6. Table 2, ASV should be uniformly used here instead of OTU.
  7. Fig. 2a, ASV should be uniformly used here instead of OTU.
  8. Lines 336-347, Generally speaking, for microbial species names, only the species name and genus name should be italicized, while other names such as phyla, classes, orders, and family names do not need to be italicized. This is a horizontal door, all names do not need to be italicized. These issues are present in many parts of the article, please carefully review and correct them.

Specific comments

  1. Line 229, Changed “community. [24]. ” to “community [24].”.
  2. Line 504, Changed “interactions [1,2,12,51]..” to “interactions [1,2,12,51].”.
  3. Line 962,  “Thiocapsa marina” should be italicized.
  4. Line 965, “Thiocapsa litoralis” should be italicized.
  5. Line 970,“Hoeflea olei” should be italicized.
  6. Line 974,“Legionella”should be italicized.
  7. Line 988,“Polynucleobacter victoriensis”should be italicized.
  8. Line 993, “Hoeflea siderophila”should be italicized.
  9. Line 996, “Silicibacter pomeroyi”and “Roseovarius nubinhibens”should be italicized.
  10. Line 1037, “Lutimaribacter litoralis ”should be italicized.

Author Response

Reviewer 2

Comment 1. The Material and Method Partneeds to be improved. There are many key methods that have not been mentioned.For examples, (1)How are sediment and water samples collected? What is their quantity? How many replicates were collected for each sample? (2)Many Physicochemical Parameters, such as NH4, and heavy metals, were mentioned in the results section (Table 1), but the author did not describe them in the methodology.

Response 1. We sincerely appreciate the reviewer's valuable feedback regarding methodological clarity. We have comprehensively revised the Materials and Methods section to include full experimental details, addressing both sampling protocols and analytical procedures such as enhanced enhanced sampling protocol (Section 2.1 line 172-186) and detailed physicochemical analyses (Section 2.2 line 188-211).

Comment 2. Lines 174-180, These sentences can be moved to the previous section (2.1).

Response 2: We appreciate this constructive suggestion for improving the manuscript's organization. As requested, we have relocated the DNA collection and stabilization protocol to Section 2.1 (now Lines 177-182), where it logically follows the general sampling description. The revised text reads:

"For microbial community analysis, DNA was extracted from independently collected replicate samples (n = 3 per matrix). Sediment (~30 g) and water samples (~50 mL) were aseptically collected using sterile disposable spoons or spatulas (pre-packaged, single-use) and immediately transferred to sterile 100 mL screw-capped tubes containing a nucleic acid stabilizing buffer (100 mM EDTA, 100 mM Tris-HCl, 150 mM NaCl, pH 8.0)."

This reorganization groups all sampling-related information together, maintains the logical flow from field collection to preservation and preserves all critical methodological details. The original section now focuses exclusively on DNA extraction and downstream processing methods.

Comment 3. Line 197-198, Amplicon sequence variant (ASV) was used in the method, but operational taxonomic units (OTUs)(Line 292)was used in the results and discussion. Please provide a unified description.

Response 3: We sincerely appreciate the reviewer’s attention to this important detail. For consistency with the high-resolution bioinformatics approach used in our study, we have replaced all instances of operational taxonomic units (OTUs) with amplicon sequence variants (ASVs) in the Results and Discussion sections (including Line 292 and related passages). This revision ensures alignment with the methodology and current best practices in microbiome analysis.

Comment 4. The unit (mg/L or mg/Kg) indication in the table is unclear. “Mg/Kg”can be placed under sediment, and “mg/L”can be placed under water.

Response 4: We appreciate the reviewer's valuable suggestion for improving table clarity. Following this recommendation, we have revised Table 1 to explicitly indicate "mg/kg" for sediment measurements and "mg/L" for water measurements. This modification ensures proper unit attribution for each matrix.

Comment 5. Lines 292, 293, 351, 390, 396, 397, ASV should be uniformly used here instead of OTU.

Response 5: We thank the reviewer for this careful observation. We have now replaced "OTU" with "ASV" in all specified locations (lines 292, 293, 351, 390, 396, 397) to maintain terminological consistency with our analytical approach. This correction has been implemented throughout the manuscript.

Comment 6. Table 2, ASV should be uniformly used here instead of OTU.

Response 6: We have updated Table 2 by replacing "OTU" with "ASV" to maintain consistency in terminology.

Comment 7. Fig. 2a, ASV should be uniformly used here instead of OTU.

Response 7: This has been corrected accordingly in line 349

Comment 8. Lines 336-347, Generally speaking, for microbial species names, only the species name and genus name should be italicized, while other names such as phyla, classes, orders, and family names do not need to be italicized. This is a horizontal door, all names do not need to be italicized. These issues are present in many parts of the article, please carefully review and correct them.

Response 8: We have corrected the formatting by removing italics from class and phylum names in the specified paragraph (lines 336-347). This correction has been applied consistently throughout the manuscript, including in the abstract. All microbial nomenclature now adheres to standard taxonomic formatting guidelines lines

Specific comments

Comment 9: Line 229, Changed “community. [24]. ” to “community [24].”.

Response 9: this has been corrected accordingly in line 229

Comment 10: Line 504, Changed “interactions [1,2,12,51]..” to “interactions [1,2,12,51].”.

Response 10: This has been corrected accordingly in line 504

Comment 11: Line 962,  “Thiocapsa marina” should be italicized.

Comment 12: Line 965, “Thiocapsa litoralis” should be italicized.

Comment 13: Line 970,“Hoeflea olei” should be italicized.

Comment 14: Line 974,“Legionella”should be italicized.

Comment  15: Line 988,“Polynucleobacter victoriensis”should be italicized.

Comment 16: Line 993, “Hoeflea siderophila”should be italicized.

Comment 17: Line 996, “Silicibacter pomeroyi”and “Roseovarius nubinhibens”should be italicized.

Comment 18: Line 1037, “Lutimaribacter litoralis ”should be italicized.

Response 11-18: This has been corrected accordingly in then reference section

Round 2

Reviewer 1 Report

Comments and Suggestions for Authors

Dear Authors,

Comments. You have improved your manuscript but it still requires some polishing. Please, check bacterial  names that have to be in italic.

General note. There is one more abundant taxon - uncl Coxiellaceae and it is ~ Delftia. It is worth to add some info about possible role of these bacteria, though I assume that won't be an easy task... Anyway it is important finding and you can make quite simple statement about it in the text.

Tissierella - can produce sulfide from peptone, please, check for refs if necessary.

30 - ... of the family ...

45 - 46 delete "including Thiocapsa and Lutimaribacter"

50 - Sulfurimonas is not a sulfate-reducing bacterium

284 The elevated salinity in sediments (5.0 g/kg vs. 3.0 g/kg in water, p < 0.001) - please, check with Table 1.

Fig 4B - Please, provide a proof (reference) that Magnetovibrio is S-reducer. Why all sulfate-reducers are shown here as S-reducers in sediment? If sulfate was present - they would function as sulfate-reducers. Perhaps, it is better to change to "Bacteria reducing inorganic sulfur compounds" ? Please, make a real correction!

Confusion with SRB designation:

518 Dominant sulfur-reducing bacteria (SRB), detected exclusively in sediment samples.

Compare: 360 bacteria (SOB) and sulfate-reducing bacteria (SRB).

537, 548 - SRB - what is here sulfur or sulfate-reducers?

Avoid confusion SRB / SRB !

485 "only 10 unique core microbiomes", perhaps: only 10 unique core genera or taxa or ...?

670 - While Nitrosomonas, and Azoarcus, are not classical SADN organisms, recent studies suggest their potential or indirect involvement under specific conditions—such as denitrification-linked sulfur oxidation [67,68].  - better to remove this sentence. IN your manuscript you have not indicate the presence of Nitrosomonas neither Azoarcus. In ref 68 it was simply indicated that Nitrosomonas can produce nitrite by ammonia oxidation under aerobic conditions.

679 Key taxa such as unclassified Desulfobacteraceae, Geobacter, Clostridium and Desulfotignum were main denitrifiers identified that have been reported elsewhere in sulfur-rich ecosystem [62,69]. - Most of there bacteria can not be the main denitrifiers!  unclassified Desulfobacteraceae & Desulfotignum could be main sulfate - reducers! Geobacter spp. normally can not denitrify!

684 - 693 Nitrosomonas was the key ammonia-oxidizing bacteria driving the first step of nitrification identified. Nitrotoga that participates in nitrite oxidation during nitrification, often in low temperature environments was the other taxa identified in both habitats. In addition, methanotrophic genera Methylobacterium and Methyloceanibacter played critical role in methane-linked processes, coupling N and C cycles. - Please, provide evidence here for all bacteria menitioned (684 - 693) how many bacteria were found in %.

689 methanotrophic genera Methylobacterium and Methyloceanibacter played critical role - Please, check if these genera are methanotrophs.

FIg 6b. (5) sulfate reduction - to make it more clear I would suggest more precise description. For example: (5) dissimilatory sulfate reduction to sulfite.

812 - 814 Sediment-associated genera such as Desulfobacteraceae, Desulfosarcina, and Thioalkalivibrio, along with minor contributors like Dethiosulfatibacter and Desulfatiglans (Figure 6b), were key contributors to ASR pathways the sediments [12,52,53].  - 1) In Fig 6b no any genera are indicated to support this conclusion! Thioalkalivibrio as a key ASR - strange! Many bacteria can be ASReducers. Is this process important for your environment? If free (dissolved) sulfide is present in the environment is ti feasible need for microorganisms to perform ASR?

815 These sulfate-reducing bacteria (SRB) have been reported - Thioalkalivibrio & Dethiosulfatibacter - please, check if these genera could reduce sulfate.

834 - 839 These genes were primarily associated with taxa belonging to α- and γ-Proteobacteria, such as Thiocapsa, Sulfitobacter, Halochromatium, Thioclava, and Pseudorhodobacter in the water column, and Desulfobacteraceae_uc, Desulfosarcina, Desulfococcus, Desulfotignum, Desulfobacter, Sulfurimonas, Thiocapsa, Halochromatium, Sulfitobacter, and Pseudorhodobacter in sediments (Figure 6b). underscoring their universal role in DSR across these niches. - It is not possible to refer here to Fig 6b, as above mentioned, genera are not shown in Figure. Are Desulfobacteraceae_uc, Desulfosarcina, Desulfococcus, Desulfotignum, Desulfobacter associated with taxa belonging to α- and γ-Proteobacteria? Please, check this.

856 Thiocapsa contributes to nitrate reduction, integrating nitrogen cycling processes and influencing nitrogen availability in aquatic systems - you do not have an experimental proof for that in your manuscript. I would suggest as Thiocapsa spp may contribute to nitrate reduction, ... (Ref).

Author Response

Comments. You have improved your manuscript but it still requires some polishing. Please, check bacterial  names that have to be in italic.

Author's Response: We sincerely thank the reviewer for the constructive and insightful comments provided during the evaluation of our manuscript. In response, we have carefully revised and refined the manuscript to address all suggestions. Specifically, all bacterial names have been checked to ensure they are properly italicized according to scientific nomenclature standards (only genus and species names have been italicized). We have corrected all inconsistencies and ensured uniform formatting throughout the manuscript. We appreciate your attention to detail, and we believe these revisions enhance both the clarity and academic rigor of the manuscript.

General note. There is one more abundant taxon - uncl Coxiellaceae and it is ~ Delftia. It is worth to add some info about possible role of these bacteria, though I assume that won't be an easy task... Anyway it is important finding and you can make quite simple statement about it in the text.

Author's Response: We have provided a narrative on the potential role of Delftia (line 855-863) and sources of unclassified Coxiellaceae that may be linked to human bathing (balneotherapyand paleotherapy using the sulfur pool mud common in isinuka) lines 870-887.

Tissierella - can produce sulfide from peptone, please, check for refs if necessary.

Author's Response: Line 425 I have added a reference to reflect the ability of Tissierella to produce H2S

30 - ... of the family ...

Author's Response: This has been corrected accordingly

45 - 46 delete "including Thiocapsa and Lutimaribacter"

Author’s response: This has been corrected according to reviewer’s suggestion

50 - Sulfurimonas is not a sulfate-reducing bacterium

Author’s response: Thank you for this correction. We acknowledge that Sulfurimonas is not a sulfate-reducing bacterium but rather a sulfur-oxidizing chemolithoautotroph commonly associated with sulfur and thiosulfate oxidation in redox transition zones. The text has been revised accordingly to remove the misclassification and to correctly describe its ecological function.

284 The elevated salinity in sediments (5.0 g/kg vs. 3.0 g/kg in water, p < 0.001) - please, check with Table 1.

Author’s response: Thank you for pointing this out. We have cross-checked with Table 1, and the correct units are mg/kg, not g/kg. The sentence now reads: "The elevated salinity in sediments (5.0 g/kg vs. 3.0 g/kg in water, p < 0.001) suggests prolonged sulfur accumulation, likely influenced by mineral precipitation and microbial SO₃²⁻ reduction processes."

Fig 4B - Please, provide a proof (reference) that Magnetovibrio is S-reducer. Why all sulfate-reducers are shown here as S-reducers in sediment? If sulfate was present - they would function as sulfate-reducers. Perhaps, it is better to change to "Bacteria reducing inorganic sulfur compounds" ? Please, make a real correction!

Authors’s response: We appreciate the reviewer’s careful observation. We apologize for the earlier oversight; the incorrect version of Figure 4B was included in the revised manuscript. Upon verification, we confirm that Magnetovibrio is not a known sulfur reducer and has therefore been removed from the updated Figure 4B. Additionally, we agree with the reviewer’s point that the taxa represented in Figure 4B are primarily sulfate-reducing bacteria (SRB), not general sulfur reducers. To address this, we have revised the figure legend to more accurately describe the represented microbial community as “sulfate-reducing bacteria (SRB)” rather than “sulfur reducers.” This revision ensures both taxonomic and functional clarity in the figure presentation and in the text.

Confusion with SRB designation: 518 Dominant sulfur-reducing bacteria (SRB), detected exclusively in sediment samples. Compare: 360 bacteria (SOB) and sulfate-reducing bacteria (SRB). 537, 548 - SRB - what is here sulfur or sulfate-reducers? Avoid confusion SRB / SRB !

Author's Response: Thank you for pointing out the potential confusion regarding the designation of sulfur-reducing bacteria (SRB). In our manuscript, we have made a distinction between sulfate-reducing bacteria (SRB) and sulfur-oxidizing bacteria (SOB). To clarify, SRB refers specifically to sulfate-reducing bacteria, which are capable of reducing sulfate to hydrogen sulfide (H₂S) in anaerobic conditions, a key process in sulfur cycling. On the other hand, SOB refers to sulfur-oxidizing bacteria, which are involved in the oxidation of sulfur compounds, such as sulfide, to sulfate under aerobic or microaerophilic conditions. In the revised manuscript, we have made sure to specify whether we are referring to sulfate-reducing bacteria (SRB) or sulfur-oxidizing bacteria (SOB) to avoid any confusion. For instance, in lines 518 and 537, where we previously mentioned "SRB," we have ensured that the context clearly indicates that we are referring to sulfate-reducing bacteria in sediment samples. Similarly, we have reviewed and corrected any ambiguous instances where the term SRB could be misunderstood, to ensure consistency and clarity throughout the manuscript.

485 "only 10 unique core microbiomes", perhaps: only 10 unique core genera or taxa or ...?

Response:  This has been recorded to indicate the core genera in line 485

670 - While Nitrosomonas, and Azoarcus, are not classical SADN organisms, recent studies suggest their potential or indirect involvement under specific conditions—such as denitrification-linked sulfur oxidation [67,68].  - better to remove this sentence. IN your manuscript you have not indicate the presence of Nitrosomonas neither Azoarcus. In ref 68 it was simply indicated that Nitrosomonas can produce nitrite by ammonia oxidation under aerobic conditions.

Author Response: We agree with the reviewer that Nitrosomonas and Azoarcus were detected in our dataset at very low abundance <0.001%, and that their link to sulfur-driven denitrification (SADN) is not directly relevant to our findings. Furthermore, considering that the sulfur pool in our study is mainly oxic, the potential for these taxa to perform SADN under such conditions is minimal. Accordingly, the sentence has been deleted from the manuscript.

679 Key taxa such as unclassified Desulfobacteraceae, Geobacter, Clostridium and Desulfotignum were main denitrifiers identified that have been reported elsewhere in sulfur-rich ecosystem [62,69]. - Most of there bacteria can not be the main denitrifiers!  unclassified Desulfobacteraceae & Desulfotignum could be main sulfate - reducers! Geobacter spp. normally can not denitrify!

Authors Response: We agree with the reviewer that Desulfobacteraceae, Desulfotignum, Geobacter, and Clostridium are not the primary denitrifiers, with Desulfobacteraceae and Desulfotignum being mainly sulfate-reducers and Geobacter primarily involved in metal reduction. In response, we have corrected the text to report only taxa known to perform denitrification in sulfur-enriched environments. Revised Text (Lines 679–681): "Key taxa such as Sulfurimonas, Paracoccus, Pseudorhodobacter, Rhodobacter, and Halomonas were the main denitrifiers identified, consistent with previous reports in sulfur-rich ecosystems [62,69]."

684 - 693 Nitrosomonas was the key ammonia-oxidizing bacteria driving the first step of nitrification identified. Nitrotoga that participates in nitrite oxidation during nitrification, often in low temperature environments was the other taxa identified in both habitats. In addition, methanotrophic genera Methylobacterium and Methyloceanibacter played critical role in methane-linked processes, coupling N and C cycles. - Please, provide evidence here for all bacteria menitioned (684 - 693) how many bacteria were found in %.

Author's Response: We have revised the text to include the relative abundances of the mentioned bacterial taxa and clarified their functional roles based on available evidence. Revised text (lines 672–680): "Nitrosomonas (0.9516% relative abundance in sediments) was identified as the key ammonia-oxidizing bacterium driving the first step of nitrification. Nitrotoga (0.7009% relative abundance in sediments), known for nitrite oxidation during nitrification under low-temperature conditions, was also identified in both habitats. In addition, the facultative methylotrophic genera Methylobacterium (0.1953% relative abundance) and Methyloceanibacter (0.224% relative abundance) may contribute to the coupling of carbon and nitrogen cycles through the metabolism of C1 compounds (such as methanol and formaldehyde) and participation in nitrogen transformations [70]."

689 methanotrophic genera Methylobacterium and Methyloceanibacter played critical role - Please, check if these genera are methanotrophs.

Authors response: Upon careful review, we acknowledge that Methylobacterium and Methyloceanibacter are not classical methanotrophic genera. Rather, they are facultative methylotrophs that utilize C1 compounds such as methanol, but do not directly oxidize methane. Accordingly, we have revised lines 688-692 in the manuscript to accurately reflect their role "Methylobacterium and Methyloceanibacter, as facultative methylotrophic genera, may contribute to the coupling of carbon and nitrogen cycles through the metabolism of C1 compounds, although they are not traditional methanotrophs."

FIg 6b. (5) sulfate reduction - to make it more clear I would suggest more precise description. For example: (5) dissimilatory sulfate reduction to sulfite.

Authors response: this has been corrected line 738 as suggested by the reviewer

834 - 839 These genes were primarily associated with taxa belonging to α- and γ-Proteobacteria, such as Thiocapsa, Sulfitobacter, Halochromatium, Thioclava, and Pseudorhodobacter in the water column, and Desulfobacteraceae_uc, Desulfosarcina, Desulfococcus, Desulfotignum, Desulfobacter, Sulfurimonas, Thiocapsa, Halochromatium, Sulfitobacter, and Pseudorhodobacter in sediments (Figure 6b). underscoring their universal role in DSR across these niches. - It is not possible to refer here to Fig 6b, as above mentioned, genera are not shown in Figure. Are Desulfobacteraceae_uc, Desulfosarcina, Desulfococcus, Desulfotignum, Desulfobacter associated with taxa belonging to α- and γ-Proteobacteria? Please, check this.

Author's Response: The sentence line 825-829 hs been revised to reflect the correct taxonomic associations, ensuring that the genera previously mentioned, including Desulfobacteraceae_uc, Desulfosarcina, Desulfococcus, Desulfotignum, and Desulfobacter, are properly classified within the δ-Proteobacteria group, alongside those associated with α- and γ-Proteobacteria - “ These genes were primarily associated with taxa belonging to α-, γ and δ-Proteobacteria, such as Thiocapsa, Sulfitobacter, Halochromatium, Thioclava, and Pseudorhodobacter in the water column, and Desulfobacteraceae_uc, Desulfosarcina, Desulfococcus, Desulfotignum, Desulfobacter, Sulfurimonas, Thiocapsa, Halochromatium, Sulfitobacter, and Pseudorhodobacter in sediments, underscoring their universal role in DSR across these niches”

856 Thiocapsa contributes to nitrate reduction, integrating nitrogen cycling processes and influencing nitrogen availability in aquatic systems - you do not have an experimental proof for that in your manuscript. I would suggest as Thiocapsa spp may contribute to nitrate reduction, ... (Ref).

Author's response: We acknowledge that our manuscript did not include experimental validation of nitrate reduction by Thiocapsa spp. In response, we have amended the relevant text to more accurately reflect the genomic-based inference rather than experimental confirmation. The text has been revised in lines 855-866 to read: "Thiocapsa spp. may contribute to nitrate reduction, thereby integrating nitrogen cycling processes and potentially influencing nitrogen availability in aquatic ecosystems (Ref). This hypothesis is supported by genomic analysis of T. bogorovii BBS, which, although unable to grow with nitrate as a sole nitrogen source and lacking key genes associated with assimilatory nitrate reduction pathways (e.g., ferredoxin-nitrate reductase [EC:1.7.7.2], nitrate reductase (NAD(P)H) [EC:1.7.1.1–1.7.1.3], and ferredoxin-nitrite reductase [EC:1.7.7.1]), harbors genes encoding enzymes involved in dissimilatory nitrate reduction and partial denitrification. Specifically, the presence of genes coding for nitrate reductase/nitrite oxidoreductase (alpha subunit) and nitrite reductase (cytochrome c-552) suggests a genomic potential for the initial steps of nitrate reduction under anaerobic conditions. Nevertheless, further experimental investigations are warranted to confirm the functional expression and ecological relevance of these pathways."

Reviewer 2 Report

Comments and Suggestions for Authors
  1. There are still some issues that have not been modified, such as using the full name when abbreviations first appear and using abbreviations afterwards. e.g. Line 360, 302, 103, 516, 827, 882…… repeated many times. There are many similar issues, please check and modify them.
  2. Line 346, (16) is it a reference. Is the format correct?

Author Response

1. There are still some issues that have not been modified, such as using the full name when abbreviations first appear and using abbreviations afterwards. e.g. Line 360, 302, 103, 516, 827, 882…… repeated many times. There are many similar issues, please check and modify them.

Author's response: We sincerely thank the reviewer for carefully pointing out these important issues. In response, we have thoroughly reviewed the manuscript to ensure that all abbreviations are first introduced with their full names upon initial mention, followed consistently by the appropriate abbreviation thereafter. We have carefully checked and corrected all instances, including those noted (e.g., Lines 103, 302, 360, 516, 827, 882), as well as other similar occurrences throughout the manuscript.

2. Line 346, (16) is it a reference. Is the format correct?

Author's response: The cited reference has been added in the reference section 

Cole, J.K.; Peacock, J.P.; Dodsworth, J.A.; Williams, A.J.; Thompson, D.B.; Dong, H.; Wu, G.; Hedlund, B.P. Sediment microbial communities in Great Boiling Spring are controlled by temperature and distinct from water communities. ISME J. 2013, 7, 718–729, doi:10.1038/ismej.2012.157.

Reviewer 3 Report

Comments and Suggestions for Authors

The author has responded to the issues I am concerned about one by one, with only one minor issue that needs to be corrected. In the vertical axis title of Fig2a, "OTUs" should be changed to "ASVs".

Author Response

The author has responded to the issues I am concerned about one by one, with only one minor issue that needs to be corrected. In the vertical axis title of Fig2a, "OTUs" should be changed to "ASVs".

Author's response: Thank you for highlighting this issue. We have carefully reviewed and corrected the identified abbreviation inconsistencies and formatting errors in the manuscript. In addition, the necessary corrections have also been made in Figure 2, where full names are now provided before using abbreviations. We appreciate the reviewer’s attention to detail, which has helped us improve the clarity and consistency of the manuscript and associated figures.